# Faster Inference of Flow-Based Generative Models via Improved Data-Noise Coupling

**Aram Davtyan**
University of Bern
Bern, Switzerland
`aram.davtyan@unibe.ch`

**Leello Tadesse Dadi**
EPFL
Lausanne, Switzerland
`leello.dadi@epfl.ch`

**Volkan Cevher**
EPFL
Lausanne, Switzerland
`volkan.cevher@epfl.ch`

**Paolo Favaro**
University of Bern
Bern, Switzerland
`paolo.favaro@unibe.ch`

## Abstract

Conditional Flow Matching (CFM), a simulation-free method for training continuous normalizing flows, provides an efficient alternative to diffusion models for key tasks like image and video generation. The performance of CFM in solving these tasks depends on the way data is coupled with noise. A recent approach uses minibatch optimal transport (OT) to reassign noise-data pairs in each training step to streamline sampling trajectories and thus accelerate inference. However, its optimization is restricted to individual minibatches, limiting its effectiveness on large datasets. To address this shortcoming, we introduce LOOM-CFM (Looking Out Of Minibatch-CFM), a novel method to extend the scope of minibatch OT by preserving and optimizing these assignments across minibatches over training time. Our approach demonstrates consistent improvements in the sampling speed-quality trade-off across multiple datasets. LOOM-CFM also enhances distillation initialization and supports high-resolution synthesis in latent space training.

## 1 Introduction

The high-quality outputs and training stability of modern diffusion (Ho et al., 2020; Dhariwal & Nichol, 2021) and flow-based models (Lipman et al., 2022; Albergo & Vanden-Eijnden, 2023), or more generally *iterative denoising methods*, have led to their rapid adoption in nearly every area of content creation, from image (Esser et al., 2024) and video generation (Davtyan et al., 2023) to motion (Hu et al., 2023) and audio synthesis (Guan et al., 2024).

Despite the superior performance of iterative methods, they require multiple evaluations of the underlying model during inference to generate content. This requirement stems from the gradual transformation of the initial sample into the desired data point, resulting in slower operation compared to single-pass methods like GANs (Goodfellow et al., 2020).

To mitigate this drawback, recent research has focused on expediting the generation process through various strategies, including enhanced training techniques (Lee et al., 2023; Bartosh et al., 2024), model distillation (Luhman & Luhman, 2021; Song et al., 2023; Liu et al., 2023; Salimans & Ho, 2022), and sampling modifications (Dhariwal & Nichol, 2021; Lu et al., 2022; Shaul et al., 2024). Specifically, iterative models employing a probability flow ordinary differential equation (ODE) framework are promising, as minimizing the curvature of their generative trajectories can significantly reduce the number of required network evaluations, thereby accelerating sampling.

This paper explores the Conditional Flow Matching (CFM) framework and its derivatives (Lipman et al., 2022; Albergo & Vanden-Eijnden, 2023; Liu et al., 2023; Tong et al., 2023). CFM provides a straightforward, efficient, and versatile training method that is not dependent on the initial noise distribution. Although data and noise are typically sampled independently, the joint distribution of data-noise pairs, or data-noise coupling, significantly impacts the curvature of sampling trajectories

in CFM-trained models (Lee et al., 2023). Sampling these pairs most effectively follows the Optimal Transport (OT) plan (Villani et al., 2009) between the noise and data distributions (Liu et al., 2023). However, computing the OT plan for large datasets is not feasible, and while minibatch OT methods offer an approximation during training (Tong et al., 2023; Pooladian et al., 2023), their effectiveness decreases with increasing dataset size.

To overcome these limitations, we introduce a method that enhances the effectiveness of minibatch OT-based CFM. Central to our approach is the preservation and iterative refinement of noise-data pairings within each minibatch, facilitating implicit communication of local OT assignments across different minibatches, thus achieving a more accurate approximation of the global OT plan. To avoid model overfitting to static noise-data assignments, we further propose assigning multiple noise instances to each data point, selecting one randomly during each training step. We refer to our approach as LOOM-CFM, standing for Looking Out Of Minibatch-CFM. We provide a convergence analysis and demonstrate through experiments that LOOM-CFM outperforms existing methods on standard benchmarks. Additionally, LOOM-CFM serves as an effective initialization for model distillation, further enhancing inference speed, and is compatible with latent flow matching for generating higher-resolution outputs. Our contributions can be summarized as follows:

- We introduce LOOM-CFM – a novel iterative algorithm to boost the generation speed and accuracy of CFMs by optimizing the global data-noise assignments of minibatch OT;
- We prevent overfitting of fixed data-noise assignments at no computational cost by allocating multiple noise samples per data point (which artificially increases the dataset size);
- We present a convergence analysis of LOOM-CFM;
- We evaluate LOOM-CFM extensively and demonstrate its superior performance over prior work. Specifically, LOOM-CFM reduces the FID with 12 NFE by 41% on CIFAR10, 46% on ImageNet-32, and 54% on ImageNet-64 compared to minibatch OT methods.

## 2 BACKGROUND

In this section, for completeness, we first introduce the required background and the prior work, which include CFM (in Section 2.1), the existing approaches to speeding up its inference by straightening the sampling paths (in Section 2.2) and optimal transport (in Section 2.3). We discuss the issues and the drawbacks of the existing methods and then explain our approach in Section 3.1.

### 2.1 CONDITIONAL FLOW MATCHING

Generative modelling requires estimating the unknown target data distribution $p(x)$ with some parametric model $p_\theta(x)$. A conventional choice for this estimator is $p_\theta(x) = \int p_\theta(x|z)p(z)\,dz$, where $p(z)$ is a given source noise distribution (typically the standard normal distribution) and $p_\theta(x|z)$ is a learned conditional noise-to-data distribution, or a generative distribution. Often the latter is considered to be deterministic, and in that case it can be written as $p_\theta(x|z) = \delta(x - g_\theta(z))$, where $g_\theta(z)$ is a deterministic mapping from noise to data, often referred to as the *generator*, and $\delta(\cdot)$ is the Dirac delta distribution.

Recent ODE-based methods (such as denoising diffusion models (Ho et al., 2020; Song et al., 2020) or conditional flow matching (Lipman et al., 2022; Albergo & Vanden-Eijnden, 2023; Liu et al., 2023; Tong et al., 2023)) implicitly define the generator $g_\theta(z)$ through the following ODE

$$\frac{d\phi(z,t)}{dt} = v_\theta(\phi(z,t),t), \tag{1}$$

$$\phi(z,0) = z, \tag{2}$$

where $v_\theta(y,t)$ is a vector field, parameterized with a neural network. $v_\theta(y,t)$ is trained in a way that $g_\theta(z) = \phi(z,1)$ induces a valid approximation of the data distribution. One way to achieve this is by minimizing the following objective

$$\mathcal{L}_{\text{CFM}}(\theta) = \mathbb{E}_{x,z\sim p(x,z),t\sim U[0,1],\varepsilon\sim\mathcal{N}(0,\sigma^2)} \left\| v_\theta(tx + (1-t)z + \varepsilon, t) - (x - z) \right\|^2, \tag{3}$$

where $p(x,z)$ is some handcrafted coupling distribution with marginals equal to $p(x)$ and $p(z)$. A common choice of $p(x,z)$ is the so-called *independent coupling* $p(x)p(z)$.

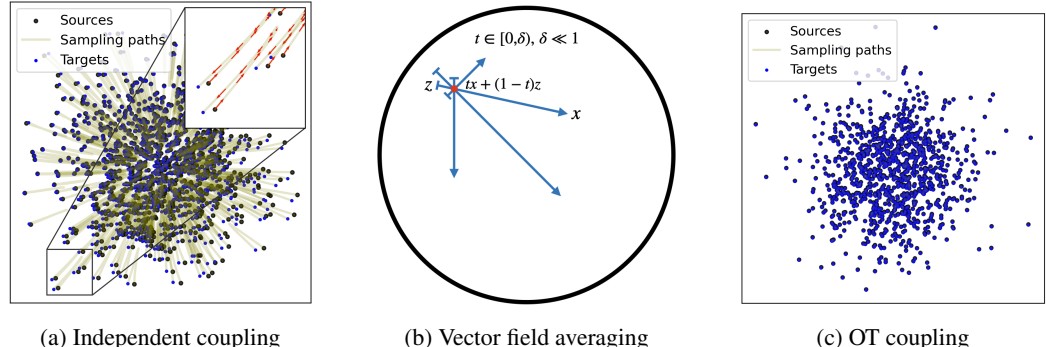

(a) Independent coupling       (b) Vector field averaging       (c) OT coupling

Figure 1: 2D Gaussian-to-Gaussian generation with CFM. This example illustrates the intuition behind the source of the curvature of the marginal sampling trajectories in the conditional flow matching framework and how optimal transport coupling could potentially solve this issue. (a) With independent coupling the sampling trajectories are highly curved. (b) This is due to the fact that in CFM the learned marginal vector field is an average over all source-to-target directions. As in the Gaussian case the distribution is symmetric around the origin, the learned vector field for small values of $t$ points towards the origin, since the target points are more likely to be sampled from the other side. (c) OT coupling resolves the curvature of the trajectories and results in the identity mapping via averaging only over the sources and targets that are close to each other.

To sample from a model trained with the CFM objective, one needs to first draw a noise sample $z$ from the source distribution and then numerically solve the ODE to obtain $g_\theta(z)$. This requires discretizing the ODE and hence calling the neural network $v_\theta(y, t)$ multiple times, which can be costly and can slow down the generation. If the trajectories of the ODE had low curvature, fewer discretization steps would be needed in order to achieve the same accuracy.

In fact, despite the convex interpolation between a single noise and data sample pair in Equation 3, being a straight line, the sampling trajectory of the ODE is actually far from straight. This can be seen even in a toy example, where the source and the target distributions are identical and are both standard normal Gaussians. Intuitively, the optimal vector field in such case should act as an identity. However, the sampling trajectories of a model trained in this setting first move towards the origin and then turn around and head back to the starting point (see Figure 1a). One might blame the convergence and numerical errors for such behavior. However, in this particular case there is a closed form analytical solution to the CFM optimization problem that is given by

$$v(y, t) = y \cdot \frac{2t - 1}{\sigma^2 + t^2 + (1 - t)^2}. \tag{4}$$

This expression clearly explains the above observation (see Appendix A for the derivation). The reason why we get such a solution is the averaging in the loss function over all directions between the source and target samples (see Figure 1b). This problem has already been noticed and pointed out in (Lee et al., 2023; Esser et al., 2024; Lee et al., 2024). Despite being just a toy example, the Gaussian to Gaussian case is actually important, as the conventional preprocessing in image generation involves normalizing images to ensure zero-mean and unit standard deviation.

## 2.2 Faster Sampling and Data Couplings

Prior work has explored different approaches to speed up the integration of the ODE trajectories. One way to achieve faster generation is to reduce the number of integration steps and hence the number of function evaluations. In order to retain accuracy with fewer integration steps straighter sampling trajectories are desired. It has been shown that the straightness of the sampling paths is highly dependent on the number of criss-crosses of the data-to-noise interpolations that is induced by the coupling distribution $p(x, z)$ (Tong et al., 2023; Lee et al., 2023). Hence, by only changing the coupling distribution, one can obtain straighter sampling trajectories of the learned marginal flow-field. In Lee et al. (2023) the authors propose to model the coupling distribution as $p(x)p(z|x)$ and to also learn the encoder or the forward process $p(z|x)$. In Bartosh et al. (2024) the forward process

is also learned and since the coupling distribution is flexible, they can additionally impose explicit constraints on the straightness of the sampling paths. However, both approaches require training of auxiliary models and incorporate additional loss functions, which complicates the training overall. In this paper we focus instead on approaches that involve only changing the coupling distribution. It has been shown that the flow obtained after optimizing Equation 3 induces a data coupling with a transport cost not larger than the cost of the initial coupling (Liu et al., 2023). More formally:

$$\mathbb{E}_{z \sim p(z)}[c(g_\theta(z), z)] \leq \mathbb{E}_{x,z \sim p(x,z)}[c(x, z)], \tag{5}$$

where $c(x, z)$ is an arbitrary convex function (*e.g.*, $\|x - z\|^2$). Thus, using the induced coupling for the second training (the *reflow* algorithm Liu et al. (2023)) tends to straighten the paths and lead to faster sampling. However, it comes with the burden of at least twice the training time. In Tong et al. (2023) and Pooladian et al. (2023) the authors noticed that one could instead use a coupling that is already optimal with respect to Equation 5 (see Figure 1c). Such a coupling is given by the so-called optimal transport plan (we introduce optimal transport in Section 2.3). Unfortunately, obtaining an exact optimal transport plan at the modern data scale is computationally infeasible. Therefore, in Tong et al. (2023) and Pooladian et al. (2023) the authors propose to approximate it via minibatch optimal transport. Tong et al. (2023) recover the soft permutation matrix with the Sinkhorn-Knopp alogirthm (Knight, 2008) and sample from it as from the joint data-noise distribution, while Pooladian et al. (2023) calculate the hard permutation matrix with the Hungarian algorithm (Kuhn, 1955) and reassign the data-noise pairs accordingly. However, the effectiveness of minibatch OT decreases with the growing sizes of the datasets. In contrast, our method, LOOM-CFM, is designed to improve minibatch OT by finding a better approximation to the global optimal transport plan via exchanging information across different minibatches. We describe our approach in Section 3.1.

## 2.3 OPTIMAL TRANSPORT

The optimal transport or Monge-Kantorovich problem refers to the search for an optimal coupling between two distributions $p(x)$ and $p(z)$, over $\mathcal{X}$ and $\mathcal{Z}$, with respect to a cost function $c(x, z)$ (Villani et al., 2009). Formally, a coupling between $p(x)$ and $p(z)$ is a distribution $p(x, z)$ over $\mathcal{X} \times \mathcal{Z}$ whose $x$ and $z$ marginals are exactly $p(x)$ and $p(z)$. In other words $\int_{\mathcal{Z}} p(x, z)dz = p(x)$ and $\int_{\mathcal{X}} p(x, z)dx = p(z)$. Among all such couplings, denoted $\Pi$, the solution to the following optimization problem

$$\inf_{p(x,z) \in \Pi} \mathbb{E}_{x,z \sim p(x,z)}[c(x, z)], \tag{6}$$

is referred to as the *optimal coupling*. Remarkably, the optimal coupling can be shown to be deterministic for distributions over $\mathbb{R}^d$ under minimal assumptions on $c$. Deterministic couplings $p(x, z)$ are those that can be expressed as the joint distribution of $(x, T(x))$ where $x \sim p(x)$ and $T : \mathcal{X} \to \mathcal{Z}$. The map $T$ *transports* $x \sim p(x)$ to $T(x) \sim p(z)$. In our setting, solving Equation 6 thus boils down to finding the optimal transport map $T$. In fact, further simplifications can be made since, in this paper, $p(x)$ will correspond to a uniform distribution over $n$ data samples, and $p(z)$ will be a uniform distribution over $n$ noise samples. Consequently, the problem considered in this paper is the optimal transport problem between two uniform distributions with finite support of equal size.

Several algorithms exist for tackling this optimization task, see Schrieber et al. (2016) for an extensive list. The problem can be cast as either a linear program or a graph-matching problem, and the most notable scheme for solving it is the Hungarian algorithm (Kuhn, 1955) which can be understood as either a primal-dual method for solving the linear program or an augmenting path approach to finding a minimum cost matching (see 3.6 in Peyré et al. (2019)). The complexity of solving Equation 6 between uniform distributions over $n$ points is $\mathcal{O}(n^3 \log(n))$. This complexity quickly becomes prohibitive for large-scale problems, which motivated the search for tractable approximations. The work of Cuturi (2013), for instance, showed that with additional regularization of 6, the resulting surrogate optimization problem can be solved with a reduced $\mathcal{O}(n^2)$ complexity but remains out of reach for modern datasets.

Alternative approximations, more relevant to our work, are randomized block-coordinate (Xie et al., 2024) and mini-batch approaches (mOT) (Fatras et al., 2021) which operate on subproblems of size $m < n$. The mini-batch approach, which is used in Pooladian et al. (2023), consists of iteratively sampling a subset of $m$ data points and $m$ noise points independently at each iteration and determining the optimal coupling with a complexity of $\mathcal{O}(m^3)$. This scheme *does not converge to the*

*optimal coupling* but to a sub-optimal, non-deterministic, averaged coupling whose bias has been analyzed in Sommerfeld et al. (2019) and greatly refined in Fatras et al. (2021). On the other hand, the work of Xie et al. (2024) proposes a randomized coordinate selection scheme to solve the linear programming formulation of 6 using a sequence of linear programs over a reduced number of variables. Convergence to optimality can be established as long as the subproblems are solved over $3n$ variables, which again remains prohibitively expensive.

Our work can be seen as an intermediate between those two approaches. At each iteration, we sample $m$ data points and their *corresponding* noise samples determined at the previous iteration, instead of the independently sampled noise points of Fatras et al. (2021). The complexity of our scheme does not exceed $\mathcal{O}(m^3)$ and is a block coordinate update scheme like Xie et al. (2024). Our method converges to a stationary transport plan, instead of an averaged one in mOT, albeit to a sub-optimal one. We describe our method in more detail in the next section.

## 3 METHOD

In this section, we first introduce our approach to finding better data-to-noise couplings (Section 3.1). Then, we propose a simple modification to the method to prevent overfitting to the fixed source (Section 3.2). Finally, we provide some implementation details (Section 3.3).

### 3.1 LOOKING OUT OF THE MINIBATCH

Suppose that we are given a set of training data points $\{x_i\}_{i\in\mathbb{N}_n}$ and noise samples $\{z_i\}_{i\in\mathbb{N}_n}$, where $\mathbb{N}_n = \{1,\dots,n\}$ and $x_i, z_i \in \mathbb{R}^d$. We aim to find a bijection $\tau^* : \mathbb{N}_n \to \mathbb{N}_n$ that represents an optimal coupling between the data and the noise sets. That is, we seek a permutation of $\mathbb{N}_n$ that satisfies

$$\tau^* = \arg\min_{\tau\in S_n} \sum_{i\in\mathbb{N}_n} c(x_i, z_{\tau(i)}), \tag{7}$$

where $c(x, z)$ is the transport cost between $x$ and $z$, and $S_n$ is the symmetric group (of degree $n$) of $\mathbb{N}_n$. In this paper, we consider the quadratic cost $c(x, z) = \|x - z\|^2$. Let $P_\tau$ be the permutation matrix corresponding to the mapping $\tau$ and let $X$ and $Z \in \mathbb{R}^{n\times d}$ be the data and noise matrices formed by stacking the corresponding samples row-wise. Then we can rewrite Equation 7 as:

$$P_{\tau^*} = \arg\min_{\tau\in S_n} \|X - P_\tau Z\|^2. \tag{OT}$$

Due to the finiteness of the feasible set, it is clear that a solution to the above problem always exists. Ideally, we would like to use $\tau^*$ as a data coupling to train CFM. That is, we would define the coupling distribution as

$$p_\tau(x, z) = \frac{1}{n} \sum_{i=1}^{n} \delta_{x_i, z_{\tau(i)}} \tag{8}$$

and use it to optimize Equation 7. However, algorithms to find $\tau^*$ usually require $O(n^3)$ running time (*e.g.*, Hungarian algorithm (Kuhn, 1955)), which makes their use infeasible at the deep learning data scale. Existing work, such as (Tong et al., 2023; Pooladian et al., 2023), approximates $\tau^*$ with a minibatch version of Equation 7. At each iteration, a new minibatch of noise is sampled and a local optimal transport is solved to assign noise samples to data within that minibatch. These assignments are used to perform the training step. Despite the simplicity of this method, it suffers from a limited scalability to larger datasets and data dimensionalities, as local assignments might be globally suboptimal. To address this issue, we propose an iterative procedure to better approximate the globally optimal assignment. First, $\{z_i\}_{i\in\mathbb{N}_n}$ are sampled and assigned to the corresponding $\{x_i\}_{i\in\mathbb{N}_n}$. At the beginning, $\tau_0$ is set to the trivial identity permutation, *i.e.*, $\tau_0(i) = i, \forall i \in \mathbb{N}_n$. At the $k$-th training iteration, a minibatch of $\{x_{n_j}\}_{j=1}^{m}$ and the correspoding $\{z_{\tau_{k-1}(n_j)}\}_{j=1}^{m}$ is sampled from $p_{\tau_{k-1}}(x, z)$. $\tau_{k-1}$ is then locally updated to ensure the optimality of Equation 7 restricted to the minibatch. That is, we find another permutation $\omega_k$ that acts locally on the current minibatch (*i.e.*, $\omega_k(i) = i$, if $i \notin \{\tau_{k-1}(n_j)\}_{j=1}^{m}$) such that the following objective is minimized

$$\omega_k = \arg\min_{\omega\in S_m} \sum_{i=1}^{m} c\left(x_{n_i}, z_{\omega(\tau_{k-1}(n_i))}\right). \tag{9}$$

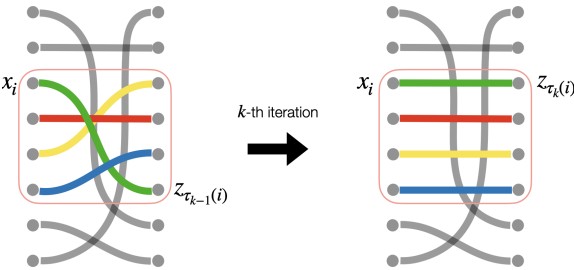

Figure 2: At each iteration a random minibatch is sampled according to the current data-to-noise assignments. Prior to taking the gradient step the current assignment is locally updated to ensure optimality w.r.t. the current minibatch. This process resembles weaving on a loom, hence the name of the method LOOM-CFM, or Looking Out Of Minibatch CFM.

---

**Algorithm 1:** LOOM-CFM

1: **Input:**
   Set of data points $\{x_i\}_{i \in \mathbb{N}_n}$,
   Set of noise samples $\{z_i\}_{i \in \mathbb{N}_n}$,
   Initial assignment $\tau_0 = \text{Id}$;
2: **for** $k$ in range$(1, T)$ **do**
3:      Sample minibatch
   $\{x_{n_j}, z_{\tau_{k-1}(n_j)}\}_{j=1}^m \sim$
   $p_{\tau_{k-1}}(x, z)$;
4:      Calculate $\omega_k$ as in Equation 9;
5:      Update $\tau_k \leftarrow \omega_k \circ \tau_{k-1}$;
6:      Take a gradient descent update
   w.r.t. Equation 7 on
   $\{x_{n_j}, z_{\tau_k(n_j)})\}_{j=1}^m$;
7: **end for**
8: **Return:** $\tau_T$

---

Hence, the update for $\tau_{k-1}$ takes the form:

$$\tau_k = \omega_k \circ \tau_{k-1}. \tag{10}$$

Finally, the training step is performed using the updated assignments from Equation 9, which are then saved for later iterations. Formally the method is described in Algorithm 1 and an illustration is provided in Figure 2.

Notice that in contrast to the prior work, the solution to the local assignment problem is not thrown away after each step, but is instead used to update the global assignment and affects the future minibatches. This makes our approach strictly better than the prior work in terms of global optimality and approximation error.

The proposed Algorithm 1 is guaranteed to converge to a stable solution. The following theorem, whose proof can be found in Appendix B, characterizes precisely the convergence behavior of LOOM-CFM. The analysis is based on casting our scheme as a randomized cycle elimination method which converges to a stationary solution.

**Theorem 1** (Finite convergence). LOOM-CFM generates a sequence of assignments $\tau_k$ with non-increasing costs. With probability 1 over the random batch selection, the iterates converge in a finite number of steps to a final assignment $\tau_{\text{final}}$ whose associated matching $M_{\text{final}}$ has no negative alternating cycles of length less than $m$.

Like all tractable approximation schemes, our algorithm is not guaranteed to recover the optimal coupling. However, unlike the minibatch OT schemes, our method outputs a deterministic coupling, and more importantly, we show in Section 4.2 that it induces a significantly better straightening of sampling paths.

## 3.2 MULTIPLE NOISE CACHES

When the size of the dataset is not large enough, using a fixed set of noise samples as the source distribution may lead to overfitting. As a result, using new noise instances as starting points for the numerical integration of the ODE in Equation 1 may lead to samples of poor quality. To overcome this issue, we propose to store more than one assigned noise sample per data point in the dataset. At each training iteration a minibatch of data-noise pairs is obtained by first sampling data points and then randomly picking one of the assigned noises. This corresponds to artificially enlarging the dataset by duplicating the data points and does not change the underlying data distribution. Later in the paper, we refer to the number of the assigned noise samples as the number of *noise caches* and show empirically that this technique helps to prevent overfitting and allows for using new noise instances as source points at inference.

### 3.3 Speed and Memory Analysis

We use the Hungarian algorithm (Kuhn, 1955) to find locally optimal assignments at each training step, making LOOM-CFM comparable in time complexity to OT-CFM (Tong et al., 2023) and BatchOT (Pooladian et al., 2023), as it solves the same minibatch matching problem. While saving and loading assignments adds minor I/O overhead, it is offset by LOOM-CFM's faster convergence. For instance, on ImageNet-32 and -64, LOOM-CFM required only 200 and 100 epochs, compared to BatchOT's 350 and 575. To minimize disk usage, we store the random numbers generator's **seed values** instead of the assigned noise instances. This incurs a negligible disk usage overhead compared to that required for storing and loading modern datasets sizes. For example, a 1M-image dataset requires only under 4MB for the noise cache. For implementation details, see Appendix C.

## 4 Experiments

In this section, we present quantitative experiments to demonstrate the effectiveness of our method on real-world data. First, in Sec.4.1, we perform ablation studies to analyze the contribution of different components of LOOM-CFM. In Sec.4.2, we compare our approach to prior work, showing that LOOM-CFM, as designed, generates higher-quality samples with fewer integration steps. Then we illustrate how LOOM-CFM enhances the initialization of the Reflow algorithm(Liu et al., 2023), removing the need for multiple Reflow iterations. Lastly, we show that our method is compatible with training in the latent space of a pre-trained autoencoder, enabling higher-resolution synthesis.

In all experiments, the main reported metric is FID (Heusel et al., 2017) which measures the distance between the distributions of some pre-trained deep features of real and generated data. It has been proven to correlate well with human perception and established as a conventional metric for image quality. As commonly done in the literature, we report the FID on the validation set with respect to 50K generated samples. The other measure that is reported is NFE (Number of Function Evaluations) that refers to the number of integration steps used to produce the corresponding result. The combination of the FID and the NFE acts as a proxy measure for the curvature of the trajectories. Low FID values along with low NFE speak for the straightness of the sampling trajectories, as the model is able to generate high-quality images using small number of steps. For qualitative results, see Appendix E.

### 4.1 Ablations

In this section, we test different versions of our algorithm to understand the impact of each component on the final performance. All ablations are conducted for unconditional generation on CIFAR10 (Krizhevsky et al., 2009), a dataset of $32 \times 32$ resolution images from 10 classes containing 50k training and 10k validation images. The metrics are reported on the validation set.

**Training with fixed source:** We start by training a naive version of CFM, by fixing the initial random assignment between the collection of noise samples and the images. Interestingly, while the results are worse than those of LOOM-CFM (see Figure 3 and Appendix D, "Fixed source, w/o reassignments"), the FID at 4 and 8 NFE are better than those of OT-CFM (Tong et al., 2023).

**Training after convergence:** A logical variant of our approach would be to wait till the algorithm converges to the stable matching and only then start training the vector field. However, this version results in slightly worse results (see Figure 3 and Appendix D, "Train After Convergence"). This might be related to overfitting to the fixed collection of noise samples and the corresponding matching. Indeed, since the algorithm starts with a random matching, training the network from the very beginning, while the matching keeps improving, can be viewed as a soft transition between the independent coupling and the coupling induced by our method.

**Training without saving reassignments:** Next, we test the impact of updating the global matching with the local reassignments. Instead of using a fixed set of noises and exchanging the information between different minibatches, for a given minibatch of images, we always sample a new minibatch of noises from the standard Gaussian distribution and assign them to the images by calculating the optimal matching. This variant replicates the method of MFM $^{\text{w}}$/ BatchOT (Pooladian et al., 2023). While the quality of generated images with this method is comparable to the quality of LOOM-CFM at large NFE, with few sampling steps LOOM-CFM is significantly better (see Figure 3 and

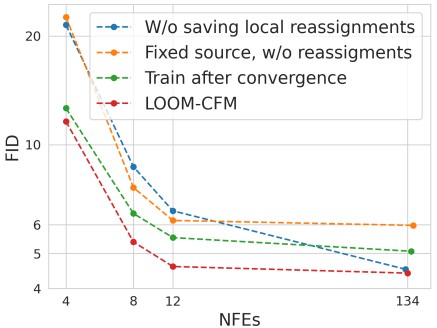

Figure 3: Ablations on different design choices on CIFAR10.

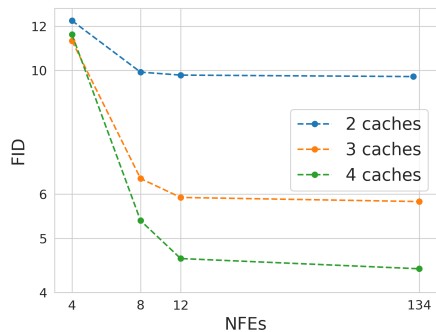

Figure 4: Ablation on the number of caches used on CIFAR10.

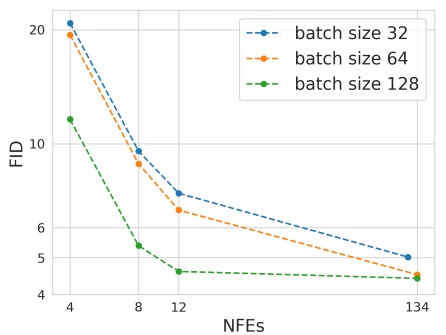

Figure 5: Ablation on the size of the minibatch used on CIFAR10.

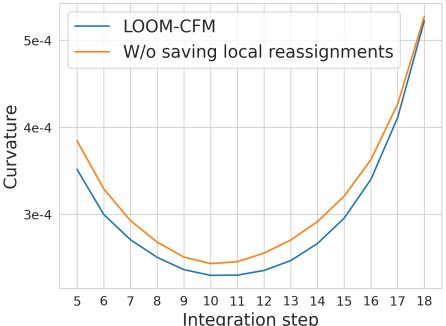

Figure 6: Curvature of the sampling trajectories for the models trained on CIFAR10. Average of 1000 trajectories is reported.

Appendix D, "W/o saving local reassignments"). Moreover, we also explicitly demonstrate that the curvature of the trajectories, measured as $1 - \bar{v}_\theta(\phi(z,t), t)^\top \bar{v}_\theta(\phi(z, t + \Delta t), t + \Delta t)$, is smaller with LOOM-CFM (see Figure 6). Here $\bar{v}_\theta$ is the normalized vector field and $t$ and $t + \Delta t$ are two consecutive integration timestamps.

**Batch size:** Further, we test LOOM-CFM with different batch sizes. As shown in Figure 5, with large NFE different batch sizes lead to similar performance. However, with small NFE larger batch size significantly overperforms the smaller ones. This indicates that the method trained with a larger batch size tends to yield straighter sampling trajectories. Nevertheless, we would like to point out that LOOM-CFM trained with batch size 32 produces on par results for 8 and 12 NFE with OT-CFM (Tong et al., 2023) that was trained with batch size 128 and outperforms it with larger batch sizes. This demonstrates the effectiveness of our caching scheme.

**Number of noise caches:** Finally, we ablate the number of noise caches used for training. Figure 4 shows that larger number of noise caches improves the performance. We found empirically, that 4 caches are enough for a dataset of size comparable to CIFAR10. We observed that the improvement diminishes with more caches, as the method becomes slower to converge and discovers worse couplings as the number of noise caches grows (see Appendix D). However, if the dataset is large enough (e.g. ImageNet (Russakovsky et al., 2015)), even a single cache may be sufficient.

## 4.2 RESULTS

**Unconditional Image Generation.** We train LOOM-CFM for unconditional generation on CI-FAR10 (Krizhevsky et al., 2009) and ImageNet32/64 (Russakovsky et al., 2015), with the results presented in Tables 1 and 2, respectively. LOOM-CFM consistently achieves better few-step FID scores compared to minibatch OT methods, such as OT-CFM (Tong et al., 2023) and MFM $^{\text{w}}/$

Table 1: Unconditional generation results on CIFAR10. (*) indicates that the reported numbers are calculated by us using the official code, as they were not reported in the corresponding papers.

| Method | Solver | CIFAR10 | |
|---|---|---|---|
| | | NFE | FID↓ |
| *ODE/SDE-based methods* | | | |
| DDPM (Ho et al., 2020) | - | 1000 | 3.17 |
| DDPM (Ho et al., 2020) | dopri5 | 274 | 7.48 |
| St. Interpolatns (Albergo & Vanden-Eijnden, 2023) | - | - | 10.27 |
| FM w/ OT (Lipman et al., 2022) | dopri5 | 142 | 6.35 |
| I-CFM (Tong et al., 2023) | euler | 100 | 4.46 |
| I-CFM (Tong et al., 2023) | euler | 1000 | 3.64 |
| I-CFM (Tong et al., 2023) | dopri5 | 146 | 3.66 |
| *Improved sampling* | | | |
| DDIM (Song et al., 2021) | - | 10 | 13.36 |
| DPM Solver-2 (Lu et al., 2022) | - | 12 | 5.28 |
| DPM Solver-3 (Lu et al., 2022) | - | 24 | 2.75 |
| RK2-BES (Shaul et al., 2024) | - | 10 | 2.73 |
| RK2-BES (Shaul et al., 2024) | - | 20 | 2.59 |
| *Rectified Flows* | | | |
| 1-Rectified Flow (Liu et al., 2023) | euler | 1 | 378 |
| + distill | euler | 1 | 6.18 |
| 2-Rectified Flow (Liu et al., 2023) | euler | 1 | 12.21 |
| + distill | euler | 1 | 4.85 |
| 3-Rectified Flow (Liu et al., 2023) | euler | 1 | 8.15 |
| + distill | euler | 1 | 5.21 |
| *Improved training for straighter trajectories* | | | |
| OT-CFM (Tong et al., 2023)* | midpoint | 4 | 15.95 |
| OT-CFM (Tong et al., 2023)* | midpoint | 8 | 9.73 |
| OT-CFM (Tong et al., 2023)* | midpoint | 12 | 7.77 |
| OT-CFM (Tong et al., 2023) | euler | 100 | 4.44 |
| OT-CFM (Tong et al., 2023) | dopri5 | 134 | 3.57 |
| Minimizing Trajectory Curvature (Lee et al., 2023) | heun | 5 | 18.74 |
| NFDM-OT (Bartosh et al., 2024) | euler | 2 | 12.44 |
| NFDM-OT (Bartosh et al., 2024) | euler | 4 | 7.76 |
| NFDM-OT (Bartosh et al., 2024) | euler | 12 | 5.20 |
| *Our results* | | | |
| LOOM-CFM (*ours*) 4 caches | midpoint | 4 | 11.60 |
| LOOM-CFM (*ours*) 4 caches | midpoint | 8 | 5.38 |
| LOOM-CFM (*ours*) 4 caches | midpoint | 12 | 4.60 |
| LOOM-CFM (*ours*) 4 caches | dopri5 | 134 | 4.41 |
| LOOM-CFM (*ours*) 4 caches + *reflow* | euler | 1 | 7.63 |
| LOOM-CFM (*ours*) 4 caches + *reflow* | midpoint | 2 | 4.49 |

BatchOT (Pooladian et al., 2023). Additionally, LOOM-CFM outperforms Lee et al. (2023) and delivers on par quality with NFDM-OT (Bartosh et al., 2024), surpassing it with 12 NFE while slightly underperforming at lower NFE. Unlike these methods, which require training additional components, LOOM-CFM optimizes the original CFM objective with a modified coupling distribution.

**Rectified Flows.** As mentioned earlier, LOOM-CFM offers improved initialization for the *reflow* algorithm (i.e., retraining the CFM using the coupling induced by the first-stage model). To validate this, we generated 1M noise-data pairs by sampling from our model trained with 4 caches on CIFAR10 and used those samples to train a new CFM model. The results, shown in Table 1 (+ *reflow*), indicate that LOOM-CFM combined with *reflow* outperforms not only 2-Rectified Flow (Liu et al., 2023) but also 3-Rectified Flow. This demonstrates that performing additional *reflows* is unnecessary as long as the first *reflow* is well-initialized.

**High-Resolution Image Synthesis.** Finally, we show that our method can be directly applied to training in the latent space of a pre-trained autoencoder, similar to the approaches in Rombach et al. (2022); Dao et al. (2023). We trained LOOM-CFM on FFHQ $256 \times 256$ (Karras et al., 2019) using a pre-trained autoencoder from Rombach et al. (2022). As shown in Table 3, LOOM-CFM achieves a lower FID score with an order of magnitude fewer NFE compared to previous methods using

Table 2: Unconditional generation results on ImageNet-32 and ImageNet-64

| Method | Solver | ImageNet-32 | | ImageNet-64 | |
|---|---|---|---|---|---|
| | | NFE | FID↓ | NFE | FID↓ |
| *ODE/SDE-based methods* | | | | | |
| St. Interpolants (Albergo & Vanden-Eijnden, 2023) | - | - | 8.49 | - | - |
| FM $^w$/ OT (Lipman et al., 2022) | dopri5 | 122 | 5.02 | 138 | 14.45 |
| *Distilled models / Dedicated solvers* | | | | | |
| *Improved training for straighter trajectories* | | | | | |
| MFM $^w$/ BatchOT (Pooladian et al., 2023) | midpoint | 4 | 17.28 | 4 | 38.45 |
| MFM $^w$/ BatchOT (Pooladian et al., 2023) | midpoint | 8 | 8.73 | 8 | 20.85 |
| MFM $^w$/ BatchOT (Pooladian et al., 2023) | midpoint | 12 | 7.18 | 12 | 18.27 |
| MFM $^w$/ Stable (Pooladian et al., 2023) | midpoint | 4 | 21.82 | 4 | 46.08 |
| MFM $^w$/ Stable (Pooladian et al., 2023) | midpoint | 8 | 9.99 | 8 | 21.36 |
| MFM $^w$/ Stable (Pooladian et al., 2023) | midpoint | 12 | 7.84 | 12 | 17.60 |
| NFDM-OT (Bartosh et al., 2024) | euler | 2 | 9.83 | 2 | 27.70 |
| NFDM-OT (Bartosh et al., 2024) | euler | 4 | 6.13 | 4 | 17.28 |
| NFDM-OT (Bartosh et al., 2024) | euler | 12 | 4.11 | 12 | 11.58 |
| *Our results* | | | | | |
| LOOM-CFM (*ours*) 1 cache | midpoint | 4 | 13.47 | 4 | 37.24 |
| LOOM-CFM (*ours*) 1 cache | midpoint | 8 | 5.08 | 8 | 11.76 |
| LOOM-CFM (*ours*) 1 cache | midpoint | 12 | 3.89 | 12 | 8.49 |
| LOOM-CFM (*ours*) 1 cache | dopri5 | 137 | 2.75 | 133 | 6.63 |

Table 3: Unconditional generation results on FFHQ 256×256

| Method | Solver | FFHQ-256 | |
|---|---|---|---|
| | | NFE | FID↓ |
| *Prior Work* | | | |
| LDM (Rombach et al., 2022) | - | 50 | 4.98 |
| ImageBART (Esser et al., 2021) | - | 3 | 9.57 |
| Latent Flow Matching (Dao et al., 2023) (ADM) | dopri5 | 84 | 8.07 |
| Latent Flow Matching (Dao et al., 2023) (DiT L/2) | dopri5 | 88 | 4.55 |
| *Our results* | | | |
| LOOM-CFM (*ours*) 4 caches (ADM) | midpoint | 2 | 14.89 |
| LOOM-CFM (*ours*) 4 caches (ADM) | midpoint | 4 | 6.76 |
| LOOM-CFM (*ours*) 4 caches (ADM) | midpoint | 8 | 5.50 |
| LOOM-CFM (*ours*) 4 caches (ADM) | dopri5 | 77 | 4.77 |

the same model architecture. This highlights the compatibility of our method with conventional techniques for high-resolution synthesis and paves the way for exploring its application to high-resolution and large-scale datasets.

## 5 CONCLUSIONS

In this paper, we introduced LOOM-CFM, a method to improve data-noise coupling in training generative models with the CFM framework. Our method is based on finding locally optimal matchings between data and noise at each minibatch. In contrast to previous work, these matchings are stored and recycled in future iterations to obtain a better data-noise assignment at the global level. Through an extensive experimental section, we established that LOOM-CFM achieves a better sampling speed-quality trade-off than prior work with no additional computational cost and negligible disk usage overhead during training. LOOM-CFM is effective and simple enough to be composed with other techniques, such as rectified flows, to further enhance the sampling speed.

## ACKNOWLEDGMENTS

This work has been supported by Swiss National Science Foundation Projects 188690 and 10001278.

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

APPENDIX

Here we provide some additional details and results that could not be included in the main paper due to the page number limit. Those include: the derivation of the optimal vector field for the Gaussian to Gaussian case (Appendix A); the proof of the finite convergence of our algorithm (Appendix B); implementation details (Appendix C); more quantitative results (Appendix D); more qualitative results (Appendix E).

## A  GAUSSIAN TO GAUSSIAN CASE

In this section, we are looking for the closed form solution for the optimal flow in the case of both source and target distributions being standard normal distributions. That is, we are aiming to finding $\hat{v}(y,t)$ such that:

$$\hat{v}(y,t) = \arg\min_{v(y,t)} \mathbb{E}_{x,z\sim\mathcal{N}(0,1),t\sim U[0,1],\varepsilon\sim\mathcal{N}(0,\sigma^2)} \left\| v(tx + (1-t)z + \varepsilon, t) - (x-z) \right\|^2, \quad (11)$$

If we write down the expectation explicitly through integrals, we obtain

$$\int_{\mathbb{R}^d} \int_{\mathbb{R}^d} \int_0^1 \int_{\mathbb{R}^d} \left\| v(y,t) - (x-z) \right\|^2 \mathcal{N}(y|tx + (1-t)z, \sigma^2 I)\mathcal{N}(z|0,I)\mathcal{N}(x|0,I) \, dydtdxdz \tag{12}$$

Let us define

$$\mathcal{L}(y,t,v) = \int_{\mathbb{R}^d} \int_{\mathbb{R}^d} \left\| v(y,t) - (x-z) \right\|^2 \mathcal{N}(y|tx + (1-t)z, \sigma^2 I) \, dxdz. \tag{13}$$

The above functional in the objective takes the form

$$\int_{\mathbb{R}^d} \int_0^1 \mathcal{L}(y,t,v(y,t)) \, dydt. \tag{14}$$

In order to find the optimal $v(y,t)$, one needs to write down the Euler-Lagrange equation (notice that $\mathcal{L}$ does not depend on the derivatives of $v$, and hence the Euler-Lagrange equation only includes the derivative w.r.t $v$)

$$\frac{\partial \mathcal{L}(y,t,v(y,t))}{\partial v} = 0, \tag{15}$$

which in this special case takes the form (here and later on we omit the integration domains for saving space and assume $\mathbb{R}^d$ everywhere)

$$\iint 2*(v(y,t) - (x-z))\mathcal{N}(y|tx + (1-t)z, \sigma^2 I)\mathcal{N}(z|0,I)\mathcal{N}(x|0,I) \, dxdz = 0. \tag{16}$$

And hence, solving for $v(y,t)$, gives

$$\hat{v}(y,t) = \frac{\iint (x-z)\mathcal{N}(y|tx + (1-t)z, \sigma^2 I)\mathcal{N}(z|0,I)\mathcal{N}(x|0,I) \, dxdz}{\iint \mathcal{N}(y|tx + (1-t)z, \sigma^2 I)\mathcal{N}(z|0,I)\mathcal{N}(x|0,I) \, dxdz} \tag{17}$$

First, let us calculate the denominator:

$$\iint \mathcal{N}(y \mid tx + (1-t)z, \sigma^2 I)\mathcal{N}(z \mid 0, I)\mathcal{N}(x \mid 0, I) \, dx \, dz = \tag{18}$$

$$\left| C_0 = \frac{1}{(2\pi)^{\frac{3d}{2}} \sigma^d} \right| \tag{19}$$

$$= C_0 \iint \exp\left(-\frac{1}{2}\left(x^\top x + z^\top z + \frac{1}{\sigma^2}\left((y - tx) - (1-t)z\right)^\top \left((y - tx) - (1-t)z\right)\right)\right) dx \, dz \tag{20}$$

$$= C_0 \iint \exp\left(-\frac{1}{2}\left(1 + \frac{(1-t)^2}{\sigma^2}\right)z^\top z + \frac{(1-t)}{\sigma^2}(y - tx)^\top z - \frac{1}{2}x^\top x - \frac{1}{2\sigma^2}(y - tx)^\top (y - tx)\right) dx \, dz \tag{21}$$

$$\left| C_1 = \frac{1}{(2\pi)^d (\sigma^2 + (1-t)^2)^{\frac{d}{2}}} \right| \tag{22}$$

$$= C_1 \int \exp\left(-\frac{1}{2}x^\top x - \frac{1}{2}\frac{1}{\sigma^2 + (1-t)^2}\left[y^\top y + t^2 x^\top x - 2t y^\top x\right]\right) dx \tag{23}$$

$$= C_1 \int \exp\left(-\frac{1}{2}\left(1 + \frac{t^2}{\sigma^2 + (1-t)^2}\right)x^\top x + \frac{t}{\sigma^2 + (1-t)^2}y^\top x - \frac{1}{2}\frac{1}{\sigma^2 + (1-t)^2}y^\top y\right) dx \tag{24}$$

$$= C_1 \frac{(2\pi)^{\frac{d}{2}}}{\left(1 + \frac{t^2}{\sigma^2 + (1-t)^2}\right)^{\frac{d}{2}}} \exp\left(\frac{1}{2}\left(\frac{t}{\sigma^2 + (1-t)^2}\right)^2 \frac{1}{1 + \frac{t^2}{\sigma^2 + (1-t)^2}}y^\top y - \frac{1}{2}\frac{1}{\sigma^2 + (1-t)^2}y^\top y\right) \tag{25}$$

$$\tag{26}$$

Here we used the well-known Gaussian integrals of the form

$$\int \exp\left(-\frac{1}{2}x^\top A x + b^\top x + c\right) dx = \sqrt{\det(2\pi A^{-1})} \exp\left(\frac{1}{2}b^\top A^{-1} b + c\right). \tag{27}$$

Finally, the last expression can be simplified to

$$(2\pi)^{-\frac{d}{2}}(\sigma^2 + t^2 + (1-t)^2)^{-\frac{d}{2}} \exp\left(-\frac{1}{2}\frac{y^\top y}{\sigma^2 + t^2 + (1-t)^2}\right), \tag{28}$$

which is the probability density function of

$$\mathcal{N}(y \mid 0, \left(\sigma^2 + t^2 + (1-t)^2\right) I). \tag{29}$$

Now let us move to the nominator, which can be rewritten as $F(y, t) - F(y, 1 - t)$, where

$$F(y, t) = \iint x \, \mathcal{N}(y|tx + (1-t)z, \sigma^2 I)\mathcal{N}(z|0, I)\mathcal{N}(x|0, I) \, dx \, dz. \tag{30}$$

Thus, we only need to calculate $F(y, t)$. Similarly to the denominator, using the well-known Gaussian integrals of the form

$$\int x \cdot \exp\left(-\frac{1}{2}x^\top x + b^\top x + c\right) dx = \left(\frac{2\pi}{a}\right)^{\frac{d}{2}} \frac{b}{a} \exp\left(\frac{b^\top b}{2a} + c\right), \tag{31}$$

we can obtain

$$F(y, t) = \frac{yt}{\sigma^2 + t^2 + (1-t)^2}\mathcal{N}(y \mid 0, \sigma^2 + t^2 + (1-t)^2). \tag{32}$$

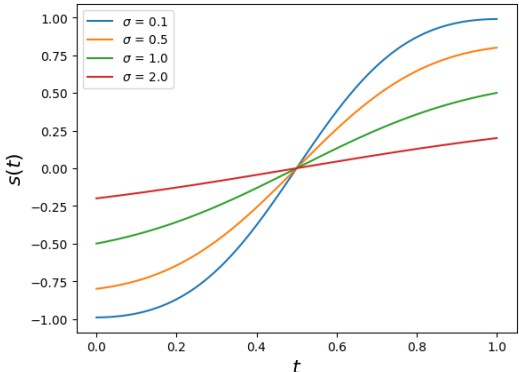

Figure 7: The form of the multiplier $s(t)$ by $y$ in the optimal vector field $v(y, t) = y \cdot s(t)$ for different values of $\sigma$.

Hence, by combining all the results, we get the optimal vector field

$$\hat{v}(y, t) = y \cdot s(t), \quad \text{where} \quad s(t) = \frac{2t - 1}{\sigma^2 + t^2 + (1 - t)^2}. \tag{33}$$

The vector field of such form is parallel to the line connecting $y$ and the origin and changes its direction halfway through time $t \in [0, 1]$ (see Figure 7). This analytical result coincides with the empirical observation of fitting the vector field as a neural network via gradient descent in the two-dimensional setting (see Section 2.1).

## B  ANALYSIS

In this section we provide a proof for the theorem established in section 3.1 and include a counter-example, where only using batch size equal to the size of the dataset could solve for the globally optimal matching (see Fig. 8).

As noted in section 2.3, OT can be understood as a minimum cost perfect matching problem in a complete bipartite graph. The assignment task is captured by the graph $\mathcal{G} = (V = X \cup Z, E)$, where $X$ is the set of nodes representing the data points and $Z$ is the set of nodes representing the noise samples. The set of edges $E = X \times E$, where the cost of an edge $(x_i, z_j) \in E$ is given by the $c_{ij} = \|x_i - z_j\|_2$, defines the complete bipartite graph (see Figure 2 for a visual representation).

A perfect matching $M$ is a set of $n$ edges in $\mathcal{G}$ that cover all the vertices. The cost of a matching is the sum of the costs of the edges. An optimal matching is one achieving the minimal cost. The optimality of a matching $M$ is entirely captured by the absence of *negative $M$-alternating cycles*.

**Definition B.1** (M-alternating cycles). Given a perfect matching $M$, any cycle in $\mathcal{G}$ given by a sequence of edges $C = (x_{i_1}, z_{i_1})(z_{i_1}, x_{i_2})(x_{i_2}, z_{i_2}) \ldots (x_{i_k}, z_{i_k})(z_{i_k}, x_{i_1})$ where for any $p \in [k]$, we have $(x_{i_p}, z_{i_p}) \in M$ is called an alternating $M$-cycle.

$M$-alternating cycles as their name suggests, pass from $X$ to $Z$ back and forth while passing through edges in $M$ to go from $X$ to $Z$. An $M$-alternating cycle is said to be negative if the sum of the costs of the edges in $M$ is larger than the cost of the edges in absent from $M$:

$$\sum_{e \in C \setminus M} c_e < \sum_{e \in C \cap M} c_e. \tag{34}$$

Consequently, for any matching $M$, if a negative $M$-alternating cycle $C$ exists, then matching $M'$ with lower cost can be constructed by taking $M' = (C \setminus M) \cup (M \setminus C)$. In other words, by swapping the *going edges* in $C$(from $X$ to $Z$) with the *return edges* (from $Z$ to $X$), we can diminish the cost thanks to Equation 34. A necessary condition for optimality is thus the absence of negative $M$-alternating cycles. The following theorem shows that it is in fact a sufficient condition.

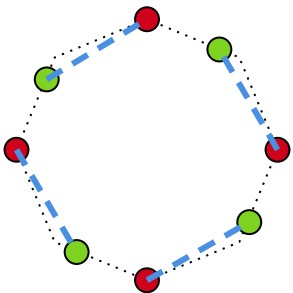

Figure 8: Counter-example showing the existence of stationary sub-optimal matchings. The data points (red) are placed at every other vertex of a regular polygon, the noise points (green) are slightly offset in a counterclockwise fashion from the vertices of the polygon. The optimal assignment matches each data points to the following noise point going in the clockwise direction. The assignment represented in blue is sub-optimal but it is stationary for any batch size $m < n$.

**Proposition B.1** (Thm 2.2 (Roughgarden, 2016)). *A matching $M$ is optimal if and only if there are no negative $M$-alternating cycles.*

Given the exposition above, LOOM-CFM can be understood as an iterative negative-cycle elimination scheme. Indeed the algorithms starts from a matching $M_0$ given by the edges $(x_i, z_{\tau_0(i)})$, then at each iteration $k$, a subgraph $\mathcal{G}_k = (V_k = \{x_{n_j}\}_{j=1}^m \cup \{z_{\tau_{k-1}(n_j)}\}_{j=1}^m, E_k)$ consisting of only the $2m$ vertices sampled at step 3 in Algorithm 1 is considered. An optimal matching is computed within that subgraph and the matching $M_{k+1}$ is obtained by updating the edges involving vertices in $V_k$. As computing optimal matchings are equivalent to elimination of all negative cycles, the update of $M_k$ corresponds to the elimination of negative $M_k$-alternating cycles that are contained in $\mathcal{G}_k$. Such cycles are of length at most $m$. The process described by algorithm 1 is thus a stepwise elimination of negative $M_k$-alternating cycles of length at most $m$. This allows us to characterize the convergence of our method in the following theorem.

**Theorem 1** (Finite convergence). LOOM-CFM generates a sequence of assignments $\tau_k$ with non-increasing costs. With probability 1 over the random batch selection, the iterates converge in a finite number of steps to a final assignment $\tau_{\text{final}}$ whose associated matching $M_{\text{final}}$ has no negative alternating cycles of length less than $m$.

*Proof.* Let $c(\tau) := \sum_{i \in \mathbb{N}_n} c(x_i, z_{\tau(i)})$. Each update of LOOM-CFM (step 4 and 5 of Algorithm 1) improves the matching over a sub-graph selected at step 3. The sequence $c(\tau_k)_k$ is thus non-increasing. Moreover, for all $k$, the sequence of costs takes its values in the finite set $\{c(\tau) : \tau \in S_n\}$. It then follows that $(c(\tau_k))_k$ becomes constant after a finite number of iterations $K_{\text{final}}$ since it is a monotone sequence taking values in a finite set.

Suppose now that there existed a negative alternating cycle of length less than $m$ in the matching induced by $\tau_{K_{\text{final}}}$. At each iteration, a subgraph containing a negative cycle has a non-zero probability of being selected at step 3. Since $(c(\tau_k))_k$ is constant for all $k > K_{\text{final}}$, this implies that such a subgraph is never sampled at step 3 for all $k > K_{\text{final}}$. The probability of such a sub-graph never being sampled for all $k > K_{\text{final}}$ is 0 by a simple application of Borell-Cantelli's lemma. Consequently, the probability of $\tau_{K_{\text{final}}}$ containing a negative alternating cycle is 0. □

Although this theorem shows that LOOM-CFM converges to a stationary solution, it does not guarantee global optimality. However, as already mentioned in Section 3.1 of the main paper, this is the case for all minibatch-based methods. We exhibit the counter-example of Xie et al. (2024) (see Figure 8) showing that any sub-problem-based approach would fail to recover the optimal coupling as a full cyclic permutation of all the points is necessary to solve the problem. Despite this, the coupling induced by LOOM-CFM is more optimal than that of minibatch OT-based methods and leads to straighter sampling trajectories, as evidenced by the experiments on real data.

Table 4: ADM network architecture and training parameters of LOOM-CFM for each model.

| | CIFAR10 | ImageNet-32 | ImageNet-64 | FFHQ256 |
|---|---|---|---|---|
| Input shape | [3, 32, 32] | [3, 32, 32] | [3, 64, 64] | [4, 32, 32] |
| Channels | 128 | 128 | 192 | 256 |
| Number of Res blocks | 2 | 3 | 2 | 2 |
| Channels multipliers | [1, 2, 2, 2] | [1, 2, 2, 2] | [1, 2, 3, 4] | [1, 2, 3, 4] |
| Heads | 4 | 4 | 4 | 4 |
| Heads channels | 64 | 64 | 64 | 64 |
| Attention resolution | [16] | [16, 8] | [16] | [16, 8, 4] |
| Dropout | 0.1 | 0.1 | 0.1 | 0.1 |
| Effective batch size | 128 | 512 | 96 | 128 |
| GPUs | 4 | 4 | 4 | 4 |
| Epochs | 1000 | 200 | 100 | 500 |
| Iterations | 391k | 500k | 1334k | 273k |
| Learning rate | 0.0002 | 0.0001 | 0.0001 | 0.00002 |
| Learning rate scheduler | Constant | Constant | Constant | Constant |
| Warmup steps | 5k | 20k | 20k | 3.5k |
| EMA decay | 0.9999 | 0.9999 | 0.9999 | 0.9999 |
| Training time (hours) | 17.3 | 73.5 | 190.6 | 66.8 |
| CFM $\sigma$ | 1e-7 | 1e-7 | 1e-7 | 1e-7 |
| Number of noise caches | 4 | 1 | 1 | 4 |

## C  IMPLEMENTATION DETAILS

This section includes the implementation details of our algorithm, including model architectures, training parameters, etc.

In practice, to be comparable with the prior work, we parametrize the learned vector field with a neural network that has an improved UNet (Ronneberger et al., 2015) architecture (ADM) from Dhariwal & Nichol (2021). To make fair comparisons, we used the same model architectures and training parameters as in the prior work, when possible. See Table 4 for the network architecture configurations and training parameters. All models (except for the ablations) were trained on 4 Nvidia RTX 3090 GPUs. For the latent space models we used the pretrained autoencoder from Stable Diffusion (Rombach et al., 2022) provided at the Hugging Face model hub[1]. The training code can be found at `https://github.com/araachie/loom-cfm`.

## D  QUANTITATIVE RESULTS

In this section, we provide more quantitative evaluations of our method.

**Ablations.** For a better visual perception, the results of ablations were presented as plots in the main paper. Here we include the numbers used to build those plots (see Table 5).

**Convergence.** Throughout the training we log the number of reassignments per minibatch. We report those in Figure 9 for different batch sizes and in Figure 11 for different number of noise caches (mimicking different dataset sizes). Additionally, we also report the minibatch OT cost around the training time where the reassignments are quite rare (Figures 10 and 12). It can be seen that LOOM-CFM with different batch sizes despite starting at different number of reassignments (around the batch size, since the intial assignments are random) converges roughly in the same amount of time (see Figure 9). However, with larger batch size the method converges to a better matching, as can be inferred from the minibatch OT cost (see Figure 10). At the same time, increasing the dataset size leads to slower convergence, since the algorithm has to visit more minibatches to sort out the assignments (see Figure 11). And the minibatch OT cost behaves accordingly (see Figure 12).

---

[1]`https://huggingface.co/stabilityai/sd-vae-ft-mse-original/blob/main/vae-ft-mse-840000-ema-pruned.ckpt`

**Training time.** As for the time complexity, LOOM-CFM is comparable to OT-CFM (Tong et al., 2023) or BatchOT (Pooladian et al., 2023) as it solves the same minibatch matching problem as the prior methods. However, saving and loading the current assignments from the disk introduces a small i/o overhead. Hence, the training time of LOOM-CFM is slightly longer than that of the baselines. For instance, training for 1000 epochs on CIFAR10 with batch size 128 scattered across 4 Nvidia RTX 3090 GPUs takes 17.3 hours with LOOM-CFM compared to 13.9 hours when the assignments are not stored.

Table 5: Detailed ablation results on CIFAR10.

| Solver | NFE | FID↓ |
|---|---|---|
| *Fixed source, w/o reassignments* | | |
| midpoint | 4 | 22.57 |
| midpoint | 8 | 7.61 |
| midpoint | 12 | 6.17 |
| dopri5 | 142 | 5.98 |
| *W/o saving local reassignments* | | |
| midpoint | 4 | 21.47 |
| midpoint | 8 | 8.69 |
| midpoint | 12 | 6.56 |
| dopri5 | 131 | 4.52 |
| *Train after convergence* | | |
| midpoint | 4 | 12.62 |
| midpoint | 8 | 6.45 |
| midpoint | 12 | 5.53 |
| dopri5 | 139 | 5.07 |
| *2 caches, bs 128* | | |
| midpoint | 4 | 12.28 |
| midpoint | 8 | 9.93 |
| midpoint | 12 | 9.81 |
| dopri5 | 126 | 9.75 |
| *3 caches, bs 128* | | |
| midpoint | 4 | 11.30 |
| midpoint | 8 | 6.40 |
| midpoint | 12 | 5.92 |
| dopri5 | 134 | 5.82 |
| *4 caches, bs 128* | | |
| midpoint | 4 | 11.60 |
| midpoint | 8 | 5.38 |
| midpoint | 12 | 4.60 |
| dopri5 | 134 | 4.41 |
| *4 caches, bs 64* | | |
| midpoint | 4 | 19.39 |
| midpoint | 8 | 8.85 |
| midpoint | 12 | 6.68 |
| dopri5 | 134 | 4.51 |
| *4 caches, bs 32* | | |
| midpoint | 4 | 20.83 |
| midpoint | 8 | 9.56 |
| midpoint | 12 | 7.40 |
| dopri5 | 122 | 5.02 |

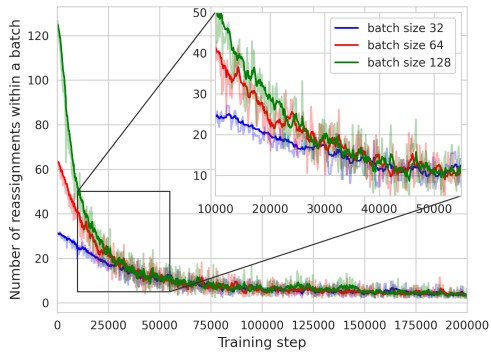

Figure 9: Number of reassignments within a minibatch along the training progress depending on the batch size.

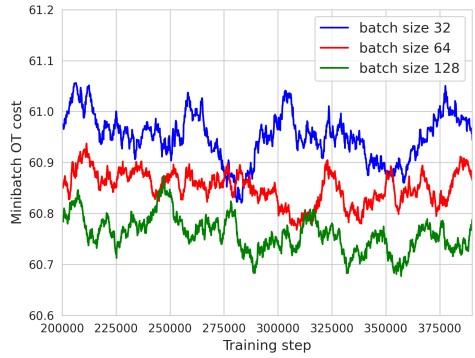

Figure 10: OT cost per minibatch along the training progress depending on the batch size. The lines corresponding to the exponential moving average of the cost are shown.

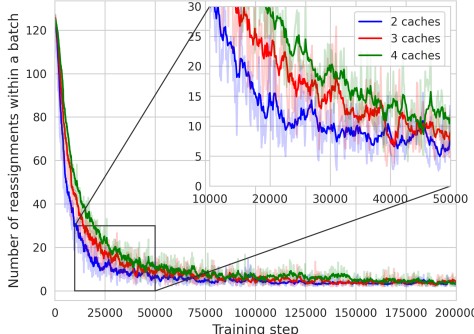 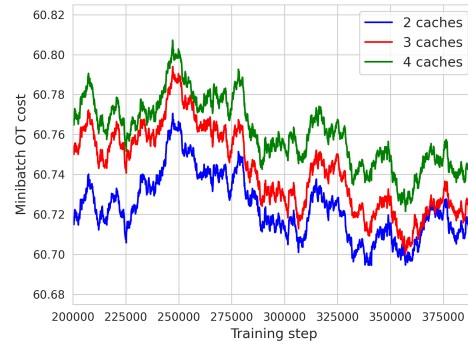

Figure 11: Number of reassignments within a minibatch along the training progress depending on the number of noise caches (the size of the dataset). The lines corresponding to the exponential moving average of the cost are shown.

Figure 12: OT cost per minibatch along the training progress depending on the number of noise caches (the size of the dataset). The lines corresponding to the exponential moving average of the cost are shown.

## E    QUALITATIVE RESULTS

In this section, we provide some qualitative results that could not be included in the main paper due to the page number limit.

**Sampling Paths.** We provide examples of sampling trajectories from our models. Starting from a randomly sampled noise instance at $t = 0$, our models iteratively denoise those via numerically integrating the ODE in Equation 1 to obtain clean images. In Figures 13, 14, 15 and 16 we show intermediate points visited by the solver at $t \leq 1$. For the model trained on CIFAR10, we also compare the sampling paths to the version of the model that does not store reassignments. For FFHQ, in Figure 17 we additionally provide samples using different number of steps in the ODE solver, resulting in different NFEs.

**Unconditional Generation.** Lastly, we provide uncurated samples from our models in Figures 18, 19, 20 and 21.

## F    ALTERNATIVE APPROACHES

In the early stages of developing LOOM-CFM, we explored various approaches to prevent overfitting to a fixed set of noise samples. Although none of these approaches yielded satisfactory outcomes, we summarize them here for clarity and completeness.

1. **Completely refreshing the noise.** We experimented with replacing all the noise samples every $N$ epochs. While intuitive, this approach caused training instability since the newly assigned noise samples did not necessarily align with the previous ones. Consequently, the network's targets could shift significantly when the noise was refreshed, impairing convergence.

2. **Gradual noise injection.** In this approach, we introduced a hyperparameter, $\phi$, to control the noise refreshing. For each data point in a minibatch, the assigned noise was replaced with a new sample with probability $\phi$. Although this method allowed for a smoother refresh, choosing an optimal $\phi$ proved to be challenging. For example, setting $\phi = 0.1$ caused approximately one third of each minibatch to be reshuffled, which weakened the effect of LOOM-CFM. In contrast, a lower value of $\phi = 0.01$ was insufficient to prevent overfitting.

3. **Interpolation between LOOM-CFM and independent coupling.** For each data point, we coupled it with its assigned noise with probability $\phi$ and with freshly sampled noise with probability $1 - \phi$. Unlike approach 2, the new noise did not replace the cached noise. We found this technique to be quite effective and the results indeed interpolated the results of LOOM-CFM and the independent coupling. We also explored making $\phi$ a function of $t$,

as the curvature of sampling paths depends on $t$ (as shown in Figure 6). Unfortunately, this hasn't lead to substantial improvements. Nevertheless, we found this approach interesting for its generality as it can interpolate any pair of coupling techniques. We leave further exploration of this method for future work.

In contrast to these more complex methods, the approach with multiple noise caches in LOOM-CFM is straightforward, as it artificially increases the dataset size and equalizes the settings for problems with small and large dataset sizes.

## G LIMITATIONS

One limitation of LOOM-CFM and other OT-based methods is that they are not directly compatible with conditional generation, especially when the conditioning signal is complex. In any conditional setup, the marginals of all label-conditional probability paths at $t = 0$ must match the source distribution. However, a naive implementation of any coupling-based method, by conditioning the model without adapting the couplings, may introduce sampling bias, as certain labels could disproportionately align with specific regions in the noise space. Although techniques like classifier(-free) guidance (Ho & Salimans, 2022) could be adapted, we leave the exploration of those extensions to future work.

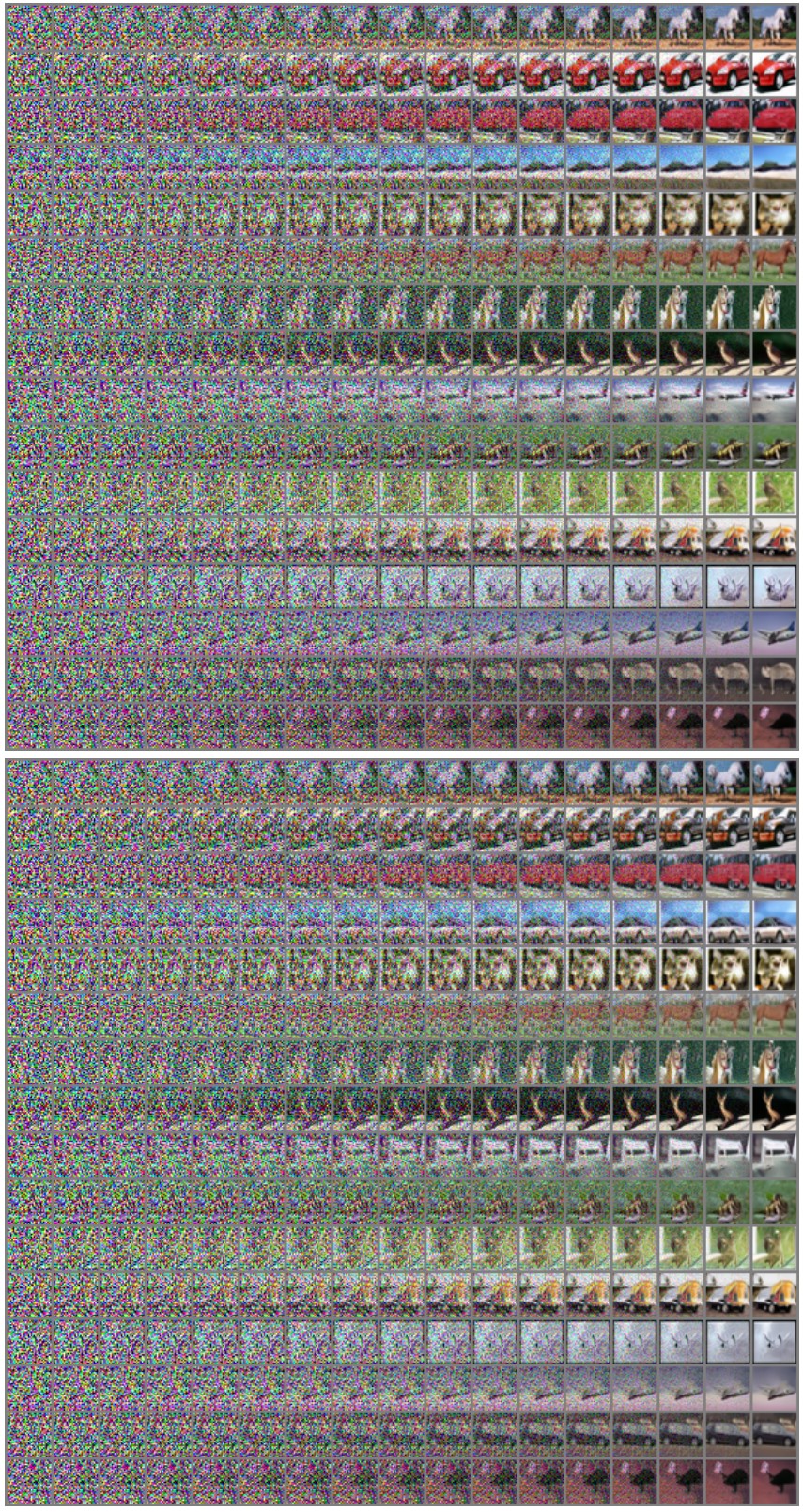

Figure 13: Sampling trajectories of LOOM-CFM (top) and the version of the algorithm that does not save the reassignments (bottom) trained on CIFAR10. The $i$-th row in both grids starts with the same initial noise. LOOM-CFM tends to produce sharper genereations and converges earlier.

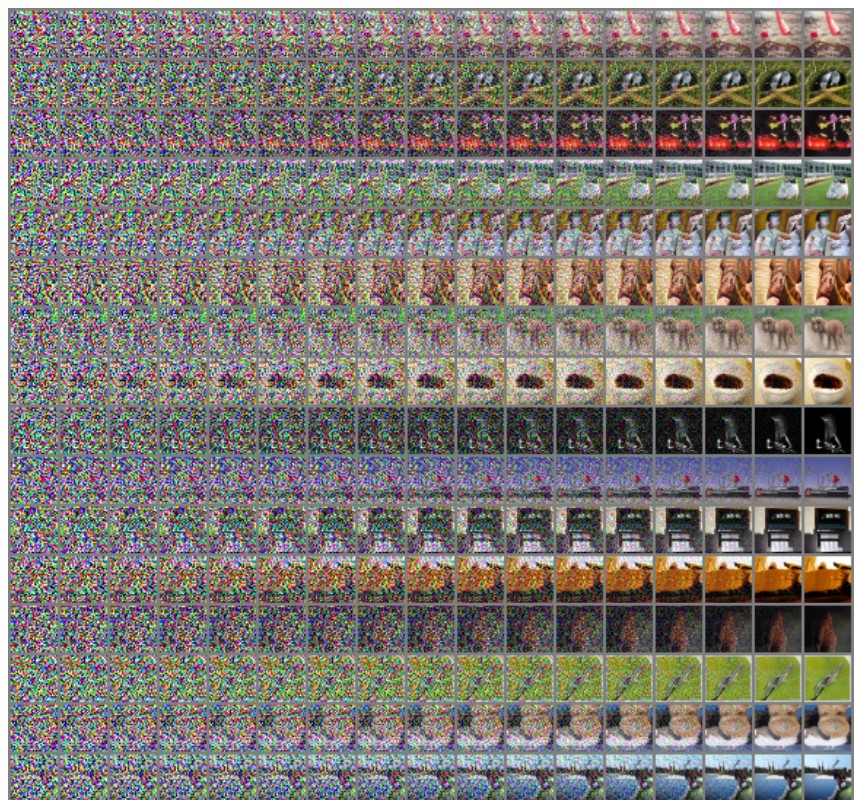

Figure 14: Sampling trajectories of the model trained on ImageNet-32.

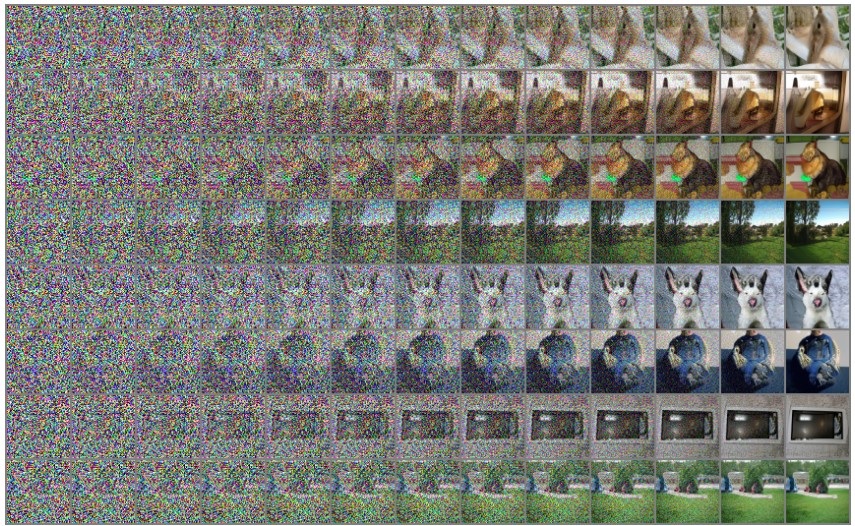

Figure 15: Sampling trajectories of the model trained on ImageNet-64.

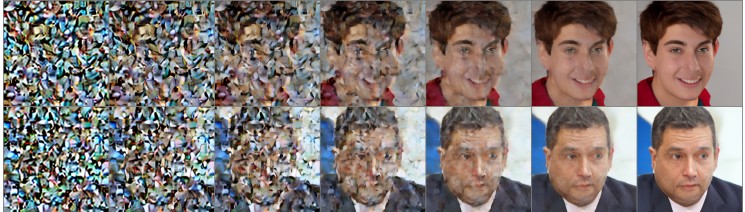

Figure 16: Sampling trajectories of the model trained on FFHQ-256.

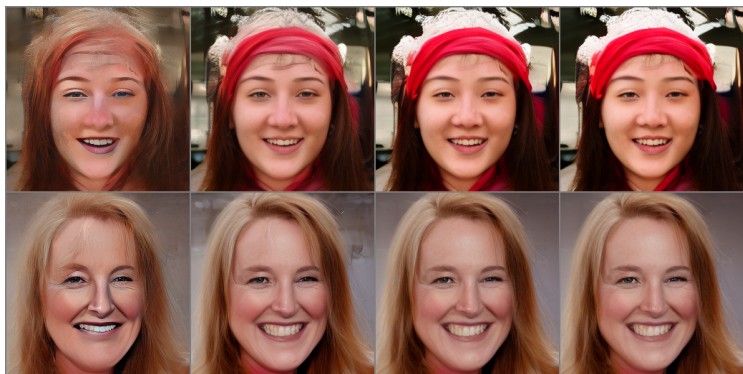

Figure 17: Samples from the FFHQ-256 model with different NFE (from left to right: 2, 4, 8 and 12 function evaluations).

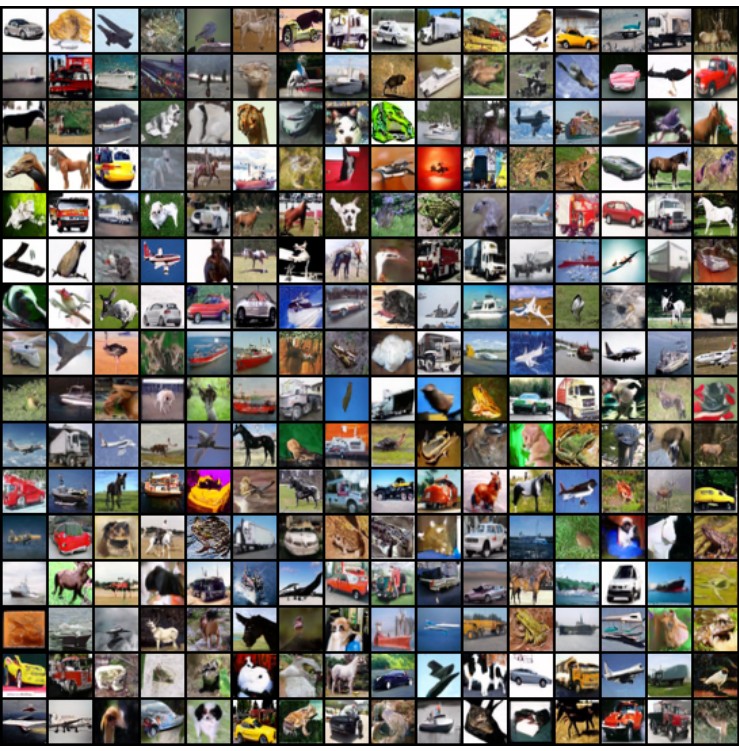

Figure 18: Uncurated samples from the model trained on CIFAR10.

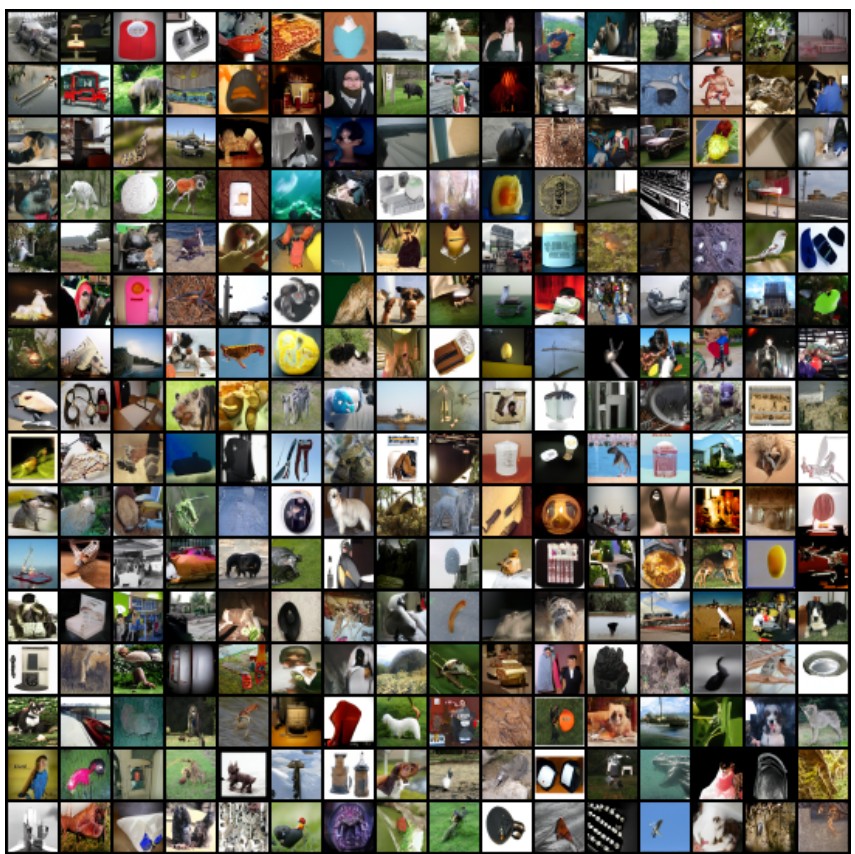

Figure 19: Uncurated samples from the model trained on ImageNet-32.

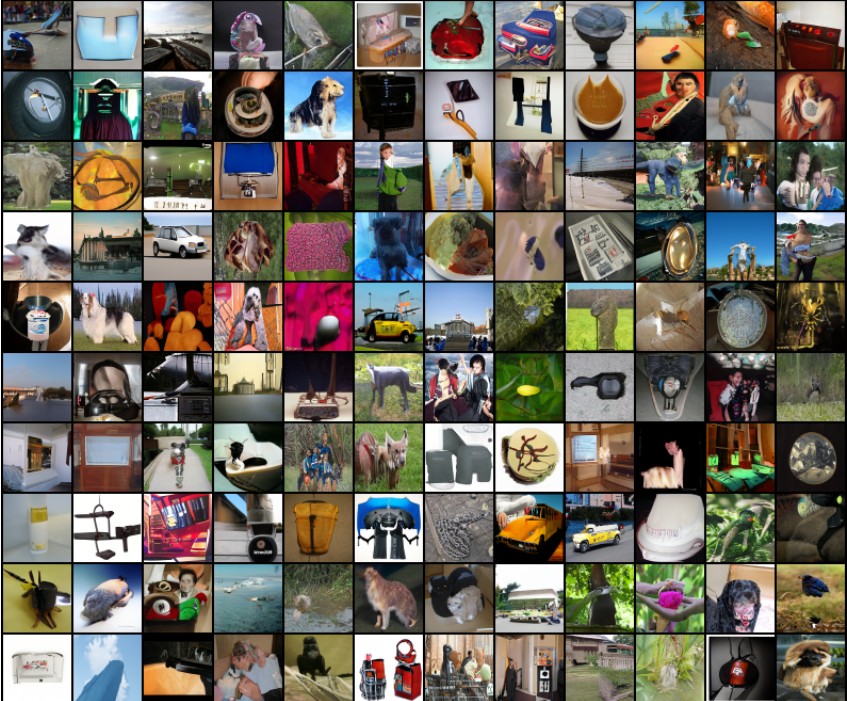

Figure 20: Uncurated samples from the model trained on ImageNet-64.

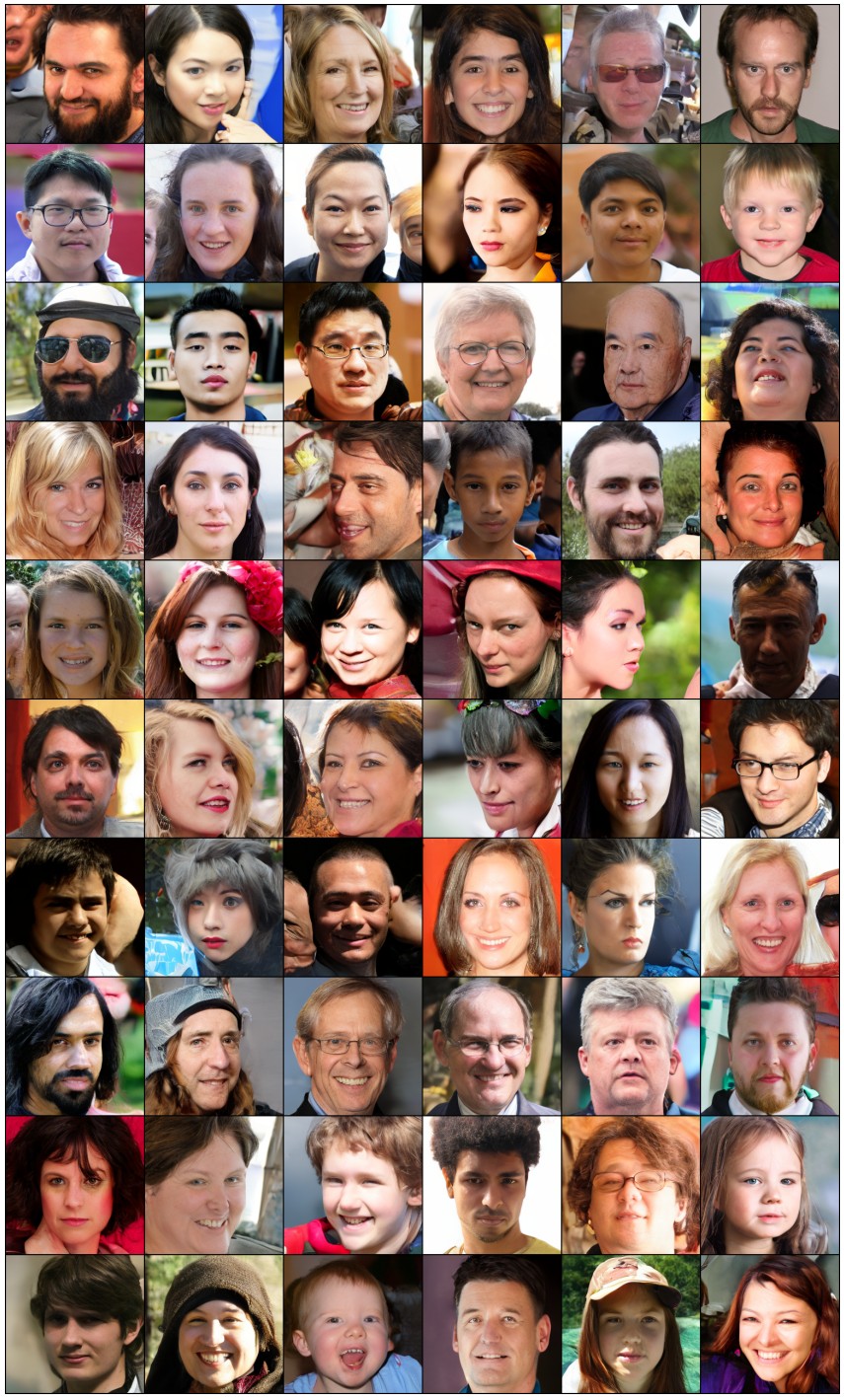

Figure 21: Uncurated samples from the model trained on FFHQ-256.

