# OpenReview forum: "Faster Inference of Flow-Based Generative Models via Improved Data-Noise Coupling"
_ICLR.cc/2025/Conference — ICLR 2025 Poster_

### Official Review · Reviewer_r9FF · 2024-10-21

**Soundness:** 3
**Presentation:** 3
**Contribution:** 3
**Rating:** 6
**Confidence:** 4

**Summary:**

The paper proposes a novel method for reassigning noise-data, which overcomes the limitations of minibatch OT reassigning. This method allows noise-data pairing beyond the confines of minibatch, reducing the curvature of the sampling trajectory and thereby accelerating the adoption process.

**Strengths:**

The paper presents a novel approach that overcomes the limitations of minibatch optimal transport (OT) by retaining coupling information within minibatches, thus facilitating the transfer of coupling information across minibatches. This advancement enables the construction of noise-data pairings that are more closely aligned with the global optimal transport framework. The proposed LOOM-CFM offers a new solution for developing superior noise-data couplings in the training of continuous flow models (CFMs). This work contributes to the application of computationally complex OT calculations in high-dimensional spaces within generative models, providing insights and references for future research in this area. Besides, the paper is well-written.

**Weaknesses:**

While LOOM-CFM demonstrates superior performance in unconditional image generation tasks compared to methods like minibatch OT, its approach of updating global pairings through optimal pairings within minibatches may not be very efficient. There exists a vast solution space for potential noise-data pairings, which poses a significant challenge. For instance, in the CIFAR-10 dataset, even when using just one cache, there are around $50000!$ possible noise-data pairing configurations. However, with a batch size of 128, each minibatch can explore a maximum of only $128!$ coupling configurations at a time. Consequently, the number of training steps required to identify the optimal pairing becomes substantially large.

**Questions:**

1. From the comparison of results between LOOM-CFM with 4 caches + reflow and LOOM-CFM with 4 caches, it can be observed that there are still many intersections in the noise-data coupling generated by LOOM-CFM. This indirectly reflects the low efficiency of the algorithm mentioned in the weaknesses. Therefore, does the issue of low efficiency in continuously exploring the global optimal pairing using minibatches impose demanding requirements on the number of training steps and training time? Could you provide the time required for training the model?
2. The meaning of the subscript $j$ in Equation 9 does not seem very clear. Could you please provide a clearer explanation?

---

> ### Author Response · Authors · 2024-11-15
> **Rebuttal by the Authors**
>
> We sincerely thank the reviewer for the valuable comments. We address the questions and clarify the issues accordingly as described in the following.
>
> **Q1: Concerns about the vast solution space.**
>
> **A1:** Thank you for the insightful comment. As explained in lines 304-307 and 912-917, our goal was not to find the globally optimal solution to discrete OT with LOOM-CFM, although that would certainly be ideal. Moreover, there are configurations of the initial pairings for which there are guarantees that the globally optimal assignment cannot be found with any of the minibatch-based approaches (see Figure 8). Thus, rather than directly outputting a globally optimal solution $\tau^*$, our method iteratively refines an initial estimate, providing an approximation that is consistently more accurate than those produced by prior minibatch-based methods and showing improved empirical metrics.
>
> Besides this, the extensive solution space does not necessarily pose a concern, as long as local updates are effective. Indeed, LOOM-CFM can be viewed as a stochastic block-coordinate descent algorithm. Such algorithms typically operate in much larger continuous solution spaces, yet converge to an optimum under specific conditions. Another example is stochastic gradient descent, widely used in optimization, that does not necessarily guarantee an optimal solution but achieves strong performance in practice.
>
> Finally, if we were to relax the optimization in Equation 7 by allowing soft assignments (i.e., permitting entries in $P_\tau$ to take any real value between 0 and 1, thereby expanding the search space even more), the problem would become a convex optimization that can be solved exactly with block-coordinate descent (as shown in, e.g., [1]). However, storing soft assignments would require $O(n^2)$ disk space, making this approach infeasible for larger datasets.
>
> **Q2: The training time of our model.**
>
> **A2:** We provided the details regarding the training time of LOOM-CFM in Appendix D, lines 968-976, as well as in Table 4. For fair comparisons, we ran our method for the same or fewer number of epochs than in the prior work. Our empirical results indicate that LOOM-CFM does not require more training time than previous approaches.
>
> **Q3: The meaning of the subscript $j$ in Equation 9.**
>
> **A3:** Thank you for spotting this! There is a typo in Equation 9: the subscript $i$ should have been used instead of $j$. We will update the paper accordingly.
>
> [1] Xie, Y., Wang, Z. \& Zhang, Z. Randomized Methods for Computing Optimal Transport Without Regularization and Their Convergence Analysis. J Sci Comput 100, 37 (2024). https://doi.org/10.1007/s10915-024-02570-w

---

> > ### Comment · Reviewer_r9FF · 2024-11-25
> >
> > Thank the authors for their responses. Most of my questions have been resolved, and I will update the rate accordingly.

---

> > > ### Author Response · Authors · 2024-11-26
> > >
> > > We would like to thank the reviewer for the thoughtful evaluation of our work and for raising the score!

---

### Official Review · Reviewer_VAZG · 2024-10-23

**Soundness:** 3
**Presentation:** 3
**Contribution:** 3
**Rating:** 6
**Confidence:** 4

**Summary:**

This paper proposes an improved method (LOOM) for computing the minibatch optimal transport coupling used to train conditional flow matching models. Unlike previous OT-CFM approaches which throw away the minibatch OT coupling after each neural net update, LOOM uses minibatch OT coupling to update the global coupling between training datasets.

**Strengths:**

**Originality** : this paper introduces an alternative (LOOM) to the Hungarian algorithm or Sinkhorn for approximating the OT coupling between two empirical measures. LOOM is suitable for conditional flow matching / minibatch flow matching in the sense that it uses local OT coupling to improve the global OT coupling.

**Clarity** : method and experiment results are clearly presented. I had no problem following the writing.

**Quality** : proposed method is supported with both theory and experiments on CIFAR10, ImageNet-32&64, and FFHQ 256x256. The experiment results support the authors' claim that conditional flow matching with LOOM yields faster flow models.

**Significance** : the paper takes a step towards scaling up OT minibatch flow matching.

**Weaknesses:**

**Weak Performance** : the only weakness of the paper stopping me from giving Accept is the weak empirical performance of the proposed method compared to relevant baselines such as [1] and [2] (which also happen to be missing from Section 4). For instance, [1] achieves 1.97 FID on CIFAR10 with 35 NFEs, whereas LOOM achieves 4.41 FID with 134 NFEs. On ImageNet-64, [2] achieves around 1.4 FID with 63 NFEs while LOOM yields 2.75 FID with 133 NFEs.

[1] Elucidating the Design Space of Diffusion-Based Generative Models

[2] Analyzing and Improving the Training Dynamics of Diffusion Models

**Questions:**

**Q1** : to what degree does LOOM accelerate convergence (in terms of number of model updates) compared to, e.g., independent coupling, Hungarian OT coupling, or Sinkhorn OT coupling?

---

> ### Author Response · Authors · 2024-11-15
> **Rebuttal by the Authors**
>
> We sincerely thank the reviewer for the positive comments on our work! We address the questions and clarify the issues accordingly as described in the following.
>
> **Q1: To what degree does LOOM-CFM accelerate model convergence compared to other coupling techniques?**
>
> **A1:** In our experiments, for fair comparison with baseline methods, we ran LOOM-CFM for the same or fewer training epochs than those used in prior work. For example, on CIFAR10, LOOM-CFM with 4 caches was trained for 1000 epochs. Notably, even within the first quarter of the training, LOOM-CFM achieved the following results that already outperform the baseline “w/o saving reassignments” (equivalent to BatchOT) for lower NFE values (see Table 5):
>
> | Solver | NFE | FID |
> | :--------: | :--------: | --------: |
> | midpoint | 4 | 14.53 |
> | midpoint | 8 | 7.46 |
> | midpoint | 12 | 6.21 |
> | dopri5 | 140 | 5.22 |
>
> *Results for LOOM-CFM, 4 caches, at 100k iterations (out of 391k)*
>
> On ImageNet-32(64) LOOM-CFM has been trained for only 200(100) epochs and outperforms BatchOT that was trained for 350(575) epochs (see Tables 2 and 4).
>
> **Q2: Relatively weaker empirical performance compared to EDM [1] and EDM2 [2].**
>
> **A2:** We did not include comparisons to [1] and [2] in section 4, as those would not be apple-to-apple, since the training settings in [1] and [2] are subtantially different from ours. To ensure apple-to-apple comparison with prior methods, we designed the experimental settings to closely match those used in baseline methods, including the same loss function, model architecture, hyperparameters, and comparable number of iterations. By isolating the data-to-noise couplings as the sole variable, we aimed to clearly highlight the advantages introduced by LOOM-CFM over previous approaches. However, we believe that LOOM-CFM’s performance could be further enhanced by combining it with additional training modifications from related works, such as those in [1] and [2], thanks to their modular nature. We illustrate this potential in our experiments by incorporating LOOM-CFM into such frameworks as Reflow and latent space training.

---

> > ### Comment · Reviewer_VAZG · 2024-11-25
> >
> > Thank you for the reply! Even though the training settings for EDM and LOOM-CFM are different, I still believe the performance gap is nontrivial. Furthermore, I think there's a large difference between claiming that LOOM-CFM can be combined with e.g., [1,2], and actually showing that it is possible. So, I will keep my original rating, which already tends towards accept.
> >
> > [1] Elucidating the Design Space of Diffusion-Based Generative Models
> >
> > [2] Analyzing and Improving the Training Dynamics of Diffusion Models

---

> > > ### Author Response · Authors · 2024-11-26
> > >
> > > Thank you for the constructive input! We understand your perspective regarding the performance gap and appreciate the suggestion to demonstrate the integration of LOOM-CFM with methods like [1, 2]. This is indeed a priority for future work, as our goal is to enhance sampling speed while maintaining high accuracy, which involves combining the best practices from both diffusion models and flow matching. Nevertheless, we believe that LOOM-CFM stands as a strong algorithm on its own, consistently achieving a better tradeoff between sampling time and quality compared to prior work under similar training settings.

---

### Official Review · Reviewer_5N5h · 2024-11-02

**Soundness:** 3
**Presentation:** 4
**Contribution:** 3
**Rating:** 6
**Confidence:** 4

**Summary:**

This paper introduced LOOM-CFM to extend the scope of minibatch OT by proposing a novel iterative algorithm to optimize the global data-noise assignments of minibatch OT. This method has consistent improvements in the sampling speed-quality trade-off and also helps enhance distillation initialization of rectified flow and supports high-resolution synthesis in latent space training.

**Strengths:**

This paper introduces an iterative approach to more accurately approximate the global optimal assignment between noise and data samples, resulting in a more precise estimation of the global OT plan compared to minibatch-CFM. It also present a convergence analysis of LOOM-CFM.

**Weaknesses:**

1. The paper lacks an analysis of the training time overhead and convergence rate of LOOM-CFM compared to minibatch-CFM. Additionally, while finite convergence is intuitive, providing a detailed analysis of the convergence rate is crucial—especially to understand the scalability of this method for large-scale text-to-image (T2I) and text-to-video (T2V) diffusion model training.

2. While the use of multiple noise caches empirically helps reduce overfitting, the one-to-many correspondence may affect the properties of probability flow. It would be valuable if the authors could provide further explanation or analysis on how multiple noise caches might or might not influence the probability flow.

**Questions:**

See weakness

---

> ### Author Response · Authors · 2024-11-15
> **Rebuttal by the Authors**
>
> We sincerely thank the reviewer for the positive comments on our work! We address the questions and clarify the issues accordingly as described in the following.
>
> **Q1: Training time overhead and convergence analysis.**
>
> **A1:** The training time overhead of LOOM-CFM is addressed in Appendix, lines 968-976. In theory, LOOM-CFM does not introduce additional computational overhead compared to other minibatch OT methods, as it performs the same operations (locally solving the optimal transport). However, a minor I/O overhead arises from loading and saving the current assignments to disk during training, which slightly impacts the training time.
>
> For the convergence analysis, we agree that theoretical convergence rate results, potentially depending on dataset and minibatch sizes, would strengthen the paper. Notably, if soft assignments were allowed (i.e., relaxing $P_\tau$ so its entries could take any value between 0 and 1), the problem in Equation 7 would become a convex optimization problem. In that case, a block-coordinate descent approach, with large enough blocks, could yield linear-time convergence (see [1] for details). However, in practice, this soft assignment approach is infeasible for large datasets due to the need for $O(n^2)$ storage and increased computational complexity. Instead, LOOM-CFM restricts updates to smaller block sizes and to permutation matrices and thus remains efficient. However, this complicates the analysis. We hypothesize that results from [1] could potentially help establish convergence bounds for LOOM-CFM, and we plan to explore this in future work.
>
> It is also important to note that LOOM-CFM is guaranteed to yield a more globally optimal coupling than other minibatch OT methods already after the 2nd epoch (when assignments start to be shared across minibatches). Moreover, empirically, we find that LOOM-CFM converges quickly enough to achieve strong results within a reasonable number of iterations.
>
> **Q2: Potential influence of multiple noise caches on the probability flow.**
>
> **A2:** As discussed in Section 3.2, storing multiple noise caches and sampling from them is equivalent to duplicating data points. Since each data point is duplicated an identical number of times, the empirical data distribution, from which points are sampled during training, is unaffected. More precisely, before adding additional noise samples per data point, the empirical data distribution is defined as:
>
> \begin{align}
>     p_{\text{data}}(x) = \frac{1}{n}\sum_{i = 1}^n \delta(x - x_i).
> \end{align}
>
> After duplicating each data points $c$ times, we obtain:
>
> \begin{align}
>     p_{\text{data}, \text{c noise caches}}(x) = \frac{1}{c \cdot n} \sum_{i = 1}^n c \cdot \delta(x - x_i) = \frac{1}{n}\sum_{i = 1}^n \delta(x - x_i) = p_{\text{data}}(x).
> \end{align}
>
> The distribution of noise samples also remains the same, as they are still drawn from the same source distribution. Since probability flow depends only on the source and target distributions, it should not be affected by the use of multiple noise caches. However, using multiple noise caches does lead to different final weights in the optimized neural network. Empirically, we observe that this approach improves generalization for smaller datasets.
>
> [1] Xie, Y., Wang, Z. \& Zhang, Z. Randomized Methods for Computing Optimal Transport Without Regularization and Their Convergence Analysis. J Sci Comput 100, 37 (2024). https://doi.org/10.1007/s10915-024-02570-w

---

> > ### Comment · Reviewer_5N5h · 2024-11-25
> > **Official Comment by Reviewer 5N5h**
> >
> > Thanks for your reply.  I still have a few follow-up questions:
> >
> > 1. In cifar10, using LOOM-CFM is 17.3 hours and not using is 13.9 hours, which is around 24% higher cost. It is not a negligible time comsumption. I am curious to know whether this overhead becomes more significant for larger datasets with higher dimensions. I also suggest including the time consumption analysis in the main body of the paper and expanding the discussion to cover different datasets for a more comprehensive evaluation.
> >
> > 2. I would like to know in the text-to-image diffusion models, how the text condition influence the coupling between noise and data. I saw the authors put it as a furture direction, but I would appreciate it if the authors could provide further insights into this point

---

> > > ### Author Response · Authors · 2024-11-26
> > >
> > > Thank you for the response! We are pleased to address the remaining questions, as outlined below:
> > >
> > > **Q3: Does training time overhead become more significant for larger datasets?**
> > >
> > > **A3:** Thank you for raising this point. Our model scales in the same way with larger datasets and higher dimensions because the single-iteration overhead depends solely on the batch size and not on data dimensionality, since the noise assignments are stored compactly as random seed values. The performance shown on the current datasets is therefore indicative of LOOM-CFM's benefits.
> > >
> > > To improve clarity, we will include the discussion on the time consumption in the main paper. Notably, LOOM-CFM achieves better performance than BatchOT while trained for significantly fewer epochs: on ImageNet-32 and -64, LOOM-CFM was trained for only 200 and 100 epochs, respectively, compared to BatchOT's 350 and 575 epochs (see Tables 2 and 4). This translates to an acceleration of up to 82% in terms of seen data samples, and thus, the performance gain of LOOM-CFM is achieved well before the additional 24% running time. For this reason, we view LOOM-CFM overall as an accelerated method compared to prior work.
> > >
> > > **Q4: How does conditioning influence the coupling between noise and data?**
> > >
> > > **A4:** Thank you for the question. LOOM-CFM has the same identical influence in the noise-data coupling from conditioning as prior coupling-based methods. In any conditional setup, the marginals of all label-conditional probability paths at $t = 0$ must match the source distribution. This is required to enable equally-accurate conditional generations from any likely noise samples. However, a naive implementation of any coupling-based method—by conditioning the model without adapting the couplings—may introduce sampling bias, as certain labels might disproportionately align with specific regions in the noise space. We will rephrase the section to clarify this. In general, addressing this challenge is an important and interesting problem that is left as future work, as noted in the conclusion.
> > >
> > > For completeness, here we provide some insights into this issue. Theoretically, this potential bias can be mitigated by restricting noise-data assignments to exchanges within the same label. While this resolves challenges in class-conditional generation, it does not directly extend to more complex conditions, such as text. However, approaches like classifier-guidance could be employed in such cases, as they operate on the unconditional probability paths. Additionally, we hypothesize that in text-to-image models, this issue is less significant due to the high diversity in the data, which naturally reduces potential biases in the noise-to-data couplings.

---

### Official Review · Reviewer_sou1 · 2024-11-04

**Soundness:** 3
**Presentation:** 3
**Contribution:** 3
**Rating:** 6
**Confidence:** 4

**Summary:**

This work investigates coupling methods for drawing paired samples of data and noise to during the training of Conditional Flow Matching (CFM) models to improve model performance. In particular, well-chosen coupling during training can make sampling paths straighter at inference time so that generation requires fewer model evaluations. Methods for achieving such a coupling are known but prohibitively expensive for large training datasets. Prior work performed the coupling only within a minibatch for Gaussian noise drawn at each step. This work improves upon this idea by using persistent noise samples whose pairings with data are gradually refined. Initially, one noise sample is drawn for each data sample at random. In each minibatch, noise and data are drawn together, and data-noise pairs are reassigned according to an efficient optimal pairing algorithm acting on the small minibatch. These new pairs are used to update the model, after which the pairs are stored until the data sample is revisited and pairings recomputed within the new batch. Multiple noise caches are used for small-scale datasets to prevent overfitting. Experimental results show significant improvement over the baseline independent coupling method and superior performance to other coupling methods, especially in the low-step regime.

**Strengths:**

* This paper has a clear focus and motivation. Coupling methods with little computational cost that can improve the performance of CFMs, especially in the low-step regime, have the potential to become widely adopted. The proposed method is scalable and the extra I/O cost seems reasonable.
* The method is clearly presented and natural. While the method remains unable to guarantee an optimal pairing (a fact which holds for any method which does not scan the entire dataset at once), it provides a richer pairing mechanism than previous minibatch approaches and remains low cost.
* The experimental results show significant improvement over the original coupling, along with strong performance relative to other few-step generation methods, and in particular among other coupling approaches.

**Weaknesses:**

* It seems somewhat limited to use only a fixed set of pre-sampled noise to train the model, in contrast to the typical approach of drawing fresh noise at every iteration. I would guess that the need for using caching for small datasets is related to the fact that the noise is not refreshed, leaving the method with a bit of a loose end. The work claims that in practice this is not an issue for large enough datasets. Nonetheless, I am curious if it is possible to define a variant of the method where noise is refreshed, eliminating the need for caches for smaller datasets.
* While the work maintains a strong focus on an important problem, actual method is fairly straightforward and heavily influence by prior work. This is not a major drawback in my view as long as the benefits that are presented in the experimental section are accurate and reproducible, because the proposed method is very easy to adopt.

**Questions:**

* How scalable is this approach? Could it be used datasets much larger than ImageNet, such as LAION-5B?
* Is the use of fixed noise an issue? Could there be a benefit from gradually refreshing the noise?

---

> ### Author Response · Authors · 2024-11-15
> **Rebuttal by the Authors**
>
> We sincerely thank the reviewer for the positive comments on our work! We address the questions and clarify the issues accordingly as described in the following.
>
> **Q1: How scalable is LOOM-CFM?**
>
> **A1:** First and foremost, LOOM-CFM consistently delivers superior couplings compared to any local minibatch OT method, regardless of dataset size. Moreover, this is achieved with negligible additional cost, making it an easy-to-implement and robust alternative for coupling noise-to-data in flow-based generative models.
>
> Some preliminary results on how LOOM-CFM scales with increasing dataset sizes can be seen in our experiments where we vary the number of noise caches. In those experiments, the batch size is fixed, while the dataset grows twice in size (from 2 noise caches to 4). Figure 11 demonstrates that while the convergence of LOOM-CFM assignments slows slightly for larger datasets, all configurations successfully converge by the end of training.
>
> Nonetheless, for an ultimate and precise conclusion on scalability of LOOM-CFM to datasets larger than ImageNet, further experiments on such large-scale datasets would be necessary, provided sufficient computational resources are available to handle such extensive training.
>
> **Q2: Refreshing the noise during training.**
>
> **A2:** Thank you for the thoughtful suggestion. Indeed, in the early stages of developing LOOM-CFM, we explored various methods to prevent overfitting to a fixed set of noise samples. Although none of these approaches yielded satisfactory outcomes, we summarize them here for clarity:
>
> 1) **Completely refreshing the noise:** We experimented with replacing all noise samples every $N$ epochs. While intuitive, this approach caused training instability since the newly assigned noise samples did not necessarily align with the previous ones. Consequently, the network’s targets could shift significantly when the noise was refreshed, impairing convergence.
> 2) **Gradual noise injection:** In this approach, we introduced a hyperparameter, $\phi$, to control noise refreshing. For each data point in a minibatch, the assigned noise was replaced with a new sample with probability $\phi$. While this method allowed for a smoother refresh, choosing an optimal $\phi$ proved challenging. For example, setting $\phi = 0.1$ led to approximately one-third of each minibatch being reshuffled, which weakened the effect of LOOM-CFM. Conversely, a lower value of $\phi=0.01$ was insufficient to prevent overfitting.
> 3) **Interpolation between LOOM-CFM and independent coupling:**  For each data point, we coupled it with its assigned noise with probability $\phi$ and with freshly sampled noise with probability $1 - \phi$. Unlike approach 2, the new noise did not replace the cached noise. We found this technique to be quite effective and the results indeed interpolated the results of LOOM-CFM and the independent coupling. We also explored making $\phi$ a function of $t$, as the curvature of sampling paths depends on $t$ (as shown in Figure 6). Unfortunately, this hasn't lead to substantial improvements. Nevertheless, we found this approach interesting for its generality, as it can interpolate any pair of coupling techniques. So we leave further exploration as future work.
>
> In contrast to these more complex methods, the approach in the paper is straightforward, as it artificially increases the dataset size and equalizes the settings for problems with small and large dataset sizes. We opted not to include all of these alternative methods in the final paper, as they diverge from the primary approach and cannot be regarded as simple ablations. However, if the reviewer believes that discussing these approaches would enhance the paper’s clarity or contribution, we would be open to incorporating a discussion of them.

---

> > ### Comment · Reviewer_sou1 · 2024-11-26
> > **Thanks for the response. I will keep my score.**
> >
> > My questions have been sufficiently addressed in the author response. I will keep my score. It is reassuring that the authors have already investigated the possibility of refreshing noise. In my view, these experiments are not essential for the main paper but could provide useful details in the appendix for future works that might follow a similar approach.

---

> > > ### Author Response · Authors · 2024-11-26
> > >
> > > Thank you for your time and valuable suggestions. We will include a discussion about these alternative approaches in the appendix.

---

### Meta-Review · Area_Chair_YBJS · 2024-12-19

**Metareview:**

This paper proposes a method to enhance mini-batch coupling in Flow Matching. It achieves this by maintaining a global bijection between noise and data, sampling mini-batches from this bijection, and updating it using local optimal transport computation. This process ensures deterministic coupling, which converges to the optimal transport in a finite number of iterations. Empirical results demonstrate that this method outperforms existing mini-batch approaches. However, it has limitations, including allowing only a finite and limited number of noise samples per data point, which can negatively impact performance. Additionally, the convergence rate is not determined. Overall, reviewers found the paper to be well-written, and the method appealing in terms of performance improvement over baselines, given the additional memory and computational costs it incurs.

**Additional Comments On Reviewer Discussion:**

No additional comments.

---

### Decision · Program_Chairs · 2025-01-22

Accept (Poster)